# Farm production diversity, household dietary diversity, and nutrition: Evidence from Uganda's national panel survey

**Haruna Sekabira**[1]*, **Zainab Nansubuga**[2], **Stanley Peter Ddungu**[2], **Lydia Nazziwa**[2]

**1** Department of Natural Resources Management, International Institute of Tropical Agriculture (IITA), Kacyiru, Rwanda, **2** Department of Agricultural Research, LADS Consult, Kampala, Uganda

* H.Sekabira@cgiar.org

**Data Availability Statement:** The data is freely publicly available at the World Bank website on the following link: https://www.worldbank.org/en/

## Abstract

Improved food security and nutrition remain a notable global challenge. Yet, food security and nutrition are areas of strategic importance regarding the United Nations' Sustainable Development Goals. The increasingly weakening global food production systems pose a threat to sustainable improved food security and nutrition. Consequently, a significant population remains chronically hungry and severely malnourished. As a remedy, farm production diversity (FPD) remains a viable pathway through which household nutrition can be improved. However, evidence is mixed, or unavailable on how FPD is associated with key nutrition indicators like household dietary diversity, energy, iron, zinc, and vitamin A (micronutrients). We use the Uganda National Panel Survey (UNPS) data for rural households to analyze differential associations of sub-components of FPD on dietary diversity, energy, and micronutrient intake. Panel data models reveal that indeed crop species count, and animal species count (sub-components of FPD) are differently associated with household dietary diversity score (HDDS), energy, and vitamin A sourced from markets. Moreover, when volumes of these nutrition outcomes were disaggregated by source (own farm vs. markets), the animal species count was only positively significantly associated with nutrition outcomes sourced from consumption of produce from own farm. Associations were insignificant for nutrition indicators sourced from markets except vitamin A. The crop species count, however, consistently showed a strong positive and significant association with energy, and all studied micronutrients sourced from own farm produce consumption, as well as those sourced from markets except Vitamin A, which was negative but insignificant. Therefore, inclusive, pro-poor, and pro-nutrition rural policy initiatives in the context of rural Uganda and similar ones, could more widely improve household nutrition through prioritizing crop species diversification on own farms because crops fetch wider nutrition gains.

## Introduction

Hunger and malnutrition remain a strong challenge in much of the developing world, despite food and nutrition security being a strategic aspect of central importance as regards the United Nations' Sustainable Development Goals [1]. The efforts of the United Nations (UN) have

programs/lsms/initiatives/lsms-ISA#8. We also provide the specific data for this particular paper in supporting information S1 File.

**Funding:** HS 2104060035 International Fund for Agricultural Development https://www.ifad.org/en/ The funders had no role in study design, data collection, and analysis, decision to publish, or preparation of the manuscript.

**Competing interests:** The authors have declared that no competing interests exist.

consistently focused on a holistic food systems approach to effectively operationalize these SDGs [1–3]. Under this approach, the full array of food systems actors that add value to food from production, collection, processing, delivery, consumption, and waste management are considered [1–5]. This broader set of activities principally aims at achieving sustainable food systems that must bring food security and nutrition to all actors [3–7] The envisioned sustainable food system must principally be sustainable economically, socially, and environmentally) [4–7]. To achieve these goals, therefore, production, distribution, processing, consumption, and other processes must be optimally prioritized [4, 5, 7]. Optimal prioritization can be achieved by optimally prioritizing activities at the different sub-systems of the food system, for instance, the farming system, and others [4, 6]. Subsequently, the needed structural changes in a food system that might ensure a food system's sustainability can be initiated from optimal changes in another sub-system like farming systems' choices of crops or animal species grown and consumed [1, 4–7]. Hence, guidance regarding a particular food system is essential to guide public policy and private investments on optimal choices across [4, 5, 7]. Generally, the UN aims to achieve a sustainable food system via the United Nations' Sustainable Development Goals (SDGs) for instance 1) ending hunger, 2) achieving food security and 3) improving nutrition for all by 2030 [1]. The UN adds that to achieve these goals, food systems must be: 1) more productive, 2) more inclusive, 3) resilient environmentally, and 4) capable of delivering healthy and nutritious foods to all [1–3]. However, since food systems are context-specific (depending on global location, culture, environment, preferences, etc.), country-specific (local) guidance is indispensable to achieving sustainable food systems [4–7]. Because of lack of such local guidance, and other inefficiencies nearly a billion people globally are chronically hungry (lack access to calories), and nearly 2 billion are exposed to micronutrient malnutrition, a matter that is largely attributed to inaccessibility to available food [1, 2, 6, 7]. However, access to food is largely determined by the availability of food and having sufficient financial resources to purchase food [8, 9]. Regrettably, access to financial resources is not guaranteed to most of the global population, hence cementing chronic poverty, hunger, and malnutrition [9, 10]. Moreover, some of the common infrastructure from which households can access food–the markets–may be rendered ineffective in availing food to the majority of the world's poorest, especially the rural poor in countries where the market infrastructure is even inadequately available [11, 12]. Therefore, farm production diversity–a key component of farming systems, remains a viable alternative to avail food to millions of the world's poorest, especially the rural majority excluded from the market infrastructure by remoteness [2, 8, 13–15].

Unfortunately, there are knowledge gaps in understanding comprehensively the nexus interconnections and linkages around associations between farm production diversity (FPD) and key nutrition outcomes [8, 16]. Such improper understanding of this nexus hinders proper policy formulation to optimally guide context-specific farming systems, thus hindering the scaling of appropriate and optimal innovations and investments against food insecurity and malnutrition, more so among vulnerable rural smallholder farmers [16]. Subsequently, such a lack of guidance and the failure to have appropriate optimal activities, render the general food systems non-inclusive [8–12]. Moreover, in most instances, evidence on how best smallholder households can access diverse diets is mixed. Some evidence points to market access being more important than diversifying farm production [17–20]. However, in the context of the least-developed countries like Uganda [21, 22], where market infrastructure is poor and rural smallholder households are trapped in poverty and remoteness, such evidence may be inapplicable. Certain evidence has documented FPD to be more important for diversity in household diets and nutrition gains [8, 11, 12, 15, 17–19, 23–29]. Hence, empirical evidence linking FPD, and nutrition outcomes is often mixed, disjointed, and incomprehensive [16, 30, 31]. For instance, none of the studies above explored the differential impacts of FPD sub-components

(animal and crop species count) on daily energy, iron, zinc, vitamin A intake, or other micronutrients. Closer efforts have been done by Muthini et al. [26], however, these covered only as far as dietary diversity without further consideration of the specific micronutrients. We contribute to this body of literature by answering the following research questions:

1. Are sub-components (crop and animal species counts) of FPD differentially associated with household dietary diversity score (HDDS), daily energy, iron, zinc, and vitamin A?

2. Which of the two FPD sub-components is associated with better nutrition gains?

Our contributions provide country-specific guidance–on potential optimal farming options in the rural Ugandan context and similar ones to reshape food production and consumption patterns [4, 5, 7]. Over the past two decades, Uganda witnessed improvements in the production of staple food crops, even though cereals like rice and wheat have also been increasingly imported but with low reliance on their consumption as staples [32–35]. The regular Ugandan diet is majorly made up of starchy roots and tubers (cassava, sweet potatoes), plantains (mostly cooking bananas), and cereals (maize, millet, sorghum). Pulses, nuts, and leafy green vegetables are usual accompaniments [32–34]. Sometimes, depending on household financial needs, food crops like bananas, sweet potatoes, pulses, and cereals can be sold for cash, alongside regular cash crops like coffee [32–35]. However, in urban areas, a nutrition transition is evolving towards imported and processed foods like rice and fast foods largely composed of meat products like chicken [32, 33, 35]. Nevertheless, the general diet especially for the poor and rural households remains needy, especially regarding foods that are rich and sufficient in micronutrients [32–35]. Therefore, food and nutrition insecurity remain persistent, due to poverty, low agricultural production, civil unrest, and climate change [32, 33]. Fortunately, in much of civil-secure rural Uganda (southern, eastern, central, western, and near northern region) the dietary energy supply from available foods, meets household energy needs, but the share of key macronutrients like lipids and proteins and micronutrients like iron, zinc, and vitamin A that are crucial for human development is grossly at the lower side of thresholds recommended by the United Nations [32–36]. For instance, globally, 2 billion persons are exposed to micronutrient malnutrition, because of low intake of vitamins and essential minerals like vitamin A, iron, and zinc [37]. These nutritional insufficiencies (micronutrient malnutrition) are caused by both the low quantities of food consumed (calories) and poor dietary diversity and quality, leading to diminished mental (learning), and physiological human growth and development, vulnerability to diseases, early deaths, and eroded economic productivity [17, 37]. About 45% (3.1 million) of deaths in children under 5 years annually are connected to micronutrient malnutrition, and in some cases lead to severe stunting and wasting [24, 37]. In 2020, 22% (149.2 million) of children below 5 years were stunted, and about 7% (45.4 million) suffered from wasting [37]. In Uganda, a third of children below five years are stunted, more so in rural areas (30%) than urban ones (24%) [38].

The human body requires calories to live, and if deprived of energy the body dies because vital organs like the heart will collapse [37, 38]. Energy is absorbed from foods consumed, and energy requirements are variant based on one's sex, age, size, and activeness [37]. In 2018 FAO, ranked Uganda 168th among 171 countries based on the volume of daily energy intake per capita, with 1,981 kilocalories, which is quite less than the African average of 2,331 (least globally) [39]. Regarding iron, globally, about 50% of anemia stands attributed to iron deficiency, ranking 9th amongst the top 26 risk components of the world's disease burdens, accounting for nearly a million deaths annually [37, 40]. The least developed world, Africa and Asia share 71% of this mortality burden, and about 50 billion USD is lost annually globally in GDP due to iron-deficiency anemia (IDA) [40]. Iron is necessary for the formation of

hemoglobin (stores and transports oxygen) [40]. Therefore, iron is so essential for life, and its deficiency severely affects children and lactating women [37, 40]. Children without sufficient iron in their life's initial thousand days experience reduced; understanding capacity, social adaptability, and language grasping skills, while foetuses would have higher probabilities of premature births, low weight, and suffocation at birth [37, 40]. For pregnant women, IDA presents higher risks of infections, and bleeding–accounting for 20% of global maternal deaths [37–40]. In Uganda, the prevalence of anemia amongst children stands at 53% [41]. Regarding zinc, about 31% of the global population suffers from zinc deficiency with rates of prevalence reaching 73% in some least developed countries, contributing to an immense burden of disease, accounting for about 58% of child deaths in Africa [37, 42]. Zinc is essential for proper human growth, reproduction, and building strong immunity against diseases, and a key anti-inflammatory mineral that is central in nearly 300 enzymes [37, 42]. The prevalence of zinc deficiency in Uganda is around 66% [28], yet zinc, iron, vitamin A, and energy have been found essential in school children's learning [43, 44]. Lastly, Vitamin A deficiency (VAD) is one of the most prevalent deficiencies globally, mostly affecting children (30% of children below 5 years) and accounting for nearly 2% of all children's deaths annually [37, 45]. VAD causes avoidable blindness in children, and VA ingestion by children via breast milk is dependent on the mother's VA status, hence VAD usually manifests early in children's lives especially in communities consuming diets that are short in vitamin A [46]. In Uganda, 28% of infant children are exposed to VAD [45], and in some regions, VAD has been found as high as 85% [28].

Therefore, understanding how food systems can substantially provide energy and these essential micronutrients is noble to attain sustainable human and economic development. With this study, we contribute to closing the knowledge gap on how farm production diversity can be tilted to ensure sufficient supplies of energy and key micronutrients, to rural Uganda where 39% of the employed population is engaged in subsistence agriculture (eat what they farm of crops and animals) [32, 35]. Regarding animal species, according to the Uganda Bureau of Statistics [35], Uganda's smallholder farmers are also engaged in livestock farming, bringing livestock numbers to about 16 million goats, 14 million cattle, 5 million sheep, 48 million poultry, and 4 million pigs. In fact, Uganda is a net exporter of live livestock especially cattle, eggs, and dairy products, most of which are produced in rural Uganda [35]. Hence, it was informative that farm production diversity is studied with disaggregated data.

The rest of the paper is organized as follows: next, we present the conceptual framework and then elaborate on the materials and methods used. We then present and discuss results while highlighting policy implications, and finally draw conclusions.

## Conceptual framework

Generally, we hypothesized that FPD bears a positive influence on food security and thus nutrition outcomes, and we diagrammatically illustrate this in Fig 1. Conceptually, following Sekabira & Nalunga [12], policies (agriculture, nutrition, or investment) influence the diversity of crops and animals species produced by farmers, thus influencing which crops or livestock species are prioritized for either direct consumption within households (own farm produce consumption pathway) or for sale to earn income and then buy food items from markets (market consumption pathway), that in the end dictate nutrition outcomes. Based on the conceptualization in Fig 1, and the empirical methodology highlighted above but elaborated later in this paper, we hypothesize that crop species count and animal species count, which are the key sub-components of FPD, associate differently with HDDS, energy, and micronutrients, and that crop species count attract stronger nutrition gains for rural households especially via the own farm consumption pathway.

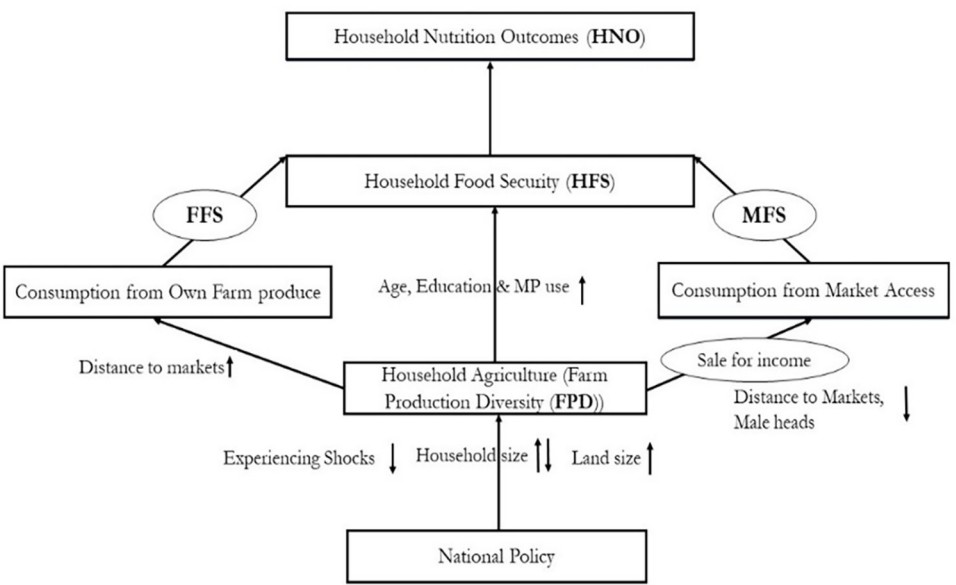

**Fig 1. Conceptual frame for farm production diversity (FPD) and nutrition nexus, adapted from Sekabira & Nalunga [12].**

## Materials and methods

### Data

**Data structure.** Generally, the data used for this study is motivated by the 50x2030 Smart Agriculture Data Initiative of the International Fund for Agricultural Development (IFAD). Specifically, we use the rural component of the Uganda National Panel Survey (UNPS) data collected by the Uganda Bureau of Statistics (UBOS), with technical support from the Living Standards Measurement Study–Integrated Surveys on Agriculture (LSMS–ISA) section of the World Bank. The sample size of the UNPS is about 3,200 households, that were previously selected and interviewed during the 2005/2006 Uganda National Household Survey (UNHS). Furthermore, the UNPS sample contains a randomly selected segment of split-off households that came into existence after the 2005/2006 UNHS. Moreover, the UNPS is both regionally and nationally representative. Each UNPS household is interviewed two times every year in an interval of six months to enhance respondent recall abilities. Data are collected and entered concurrently using computer-assisted interview applications (CAPI), installed on mobile personal computers that are operated by trained graduate enumerators. Subsequently, when data are fully cleaned and documented, they are made available to the public in a period of twelve months [47]. The UNPS has seven waves including 2009/10, 2010/11, 2011/12, 2013/14, 2015/16, 2018/19, and 2019/20. This data is collected once every year, even though the exercise is planned to be completed in two calendar years given other data management needs. We only use the latest 3 waves because these were possible to merge and analyze together given a common structure of households' identification. We had also separately worked with earlier waves on studying similar nutrition outcomes. Essentially, we built on analyses of Sekabira & Nalunga [12] and Sekabira et al. [28] who used 2009/10, 2010/2011, and 2011/2012 waves. Data are sourced from across Uganda covering the different farming systems under which different crops and livestock species are produced. Most commonly are the intensive banana and coffee farming systems around the lake shores in southern Uganda, the annual cropping and cattle farming systems in northern Uganda, the banana-coffee-cattle farming systems in

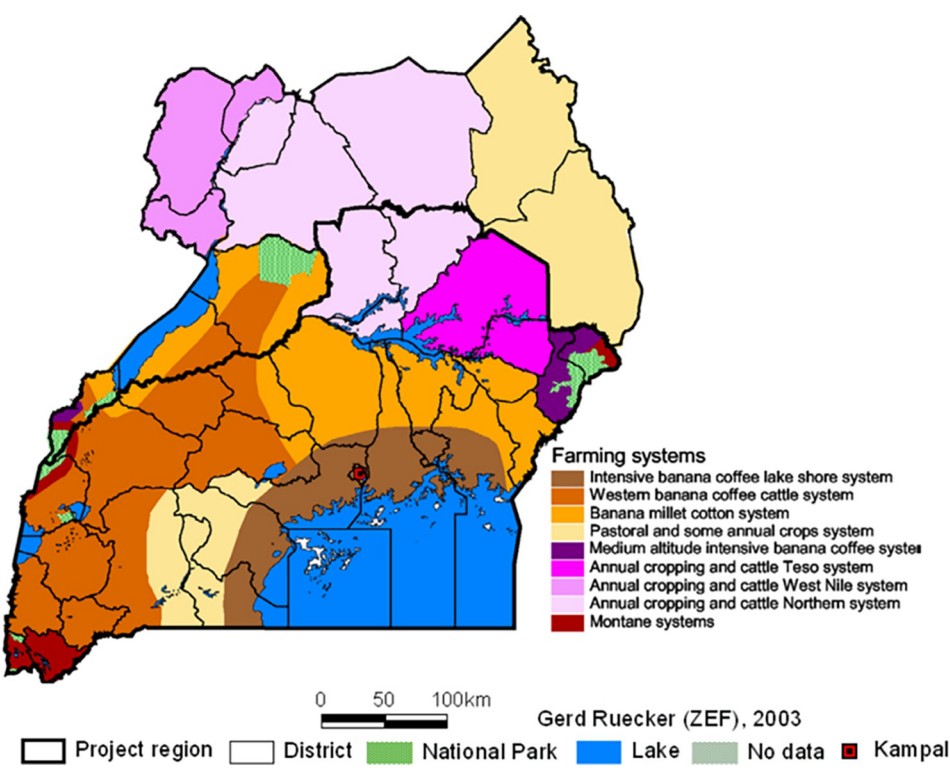

**Fig 2. Major farming systems of Uganda, as sourced from Global Yield Gap Atlas [48].**

western Uganda, and the pastoral/ annual crops farming systems in the northeast [32, 35, 48]. Details of these farming systems and geographic areas of rural Uganda where data was collected are illustrated in Fig 2, sourced from Global Yield Gap Atlas [48].

**Measurement of key variables.** *Farm production diversity (FPD)* was measured using the biodiversity index, which is a simple count of all crops and livestock produced on farm, as previously used by [11, 12, 15, 17, 49]. Therefore, before generating the biodiversity index for FPD, we generated its sub-components, the species count based on crops, and the animal species count based on livestock. Commercial but edible crops and animal species have also been considered in the calculation of FPD–since the markets consumption pathway which is instigated by market sales based on different crop and animal species grown by households is considered in this study. Moreover, some crops and animal species are used for both food and cash making purposes, depending on prevalent household needs. Examples of some of the crops used in calculation of the FDP include, rice, maize, millet, sorghum, beans, cowpeas, chickpeas, groundnuts, soya beans, sunflower, Simsim, cabbage, tomatoes, carrots, onions, pumpkins, amaranth, eggplants, cucumber, green pepper, sugarcane, potatoes, sweet potatoes, cassava, yams, oranges, pawpaw, pineapples, bananas, mangoes, avocado, passion fruit, watermelon, coffee, and ginger. Some of the animal species considered included cattle, goats, sheep, pigs, chicken, and rabbits. *Dietary diversity* was measured using the aggregated food index which measures the sum of food groups (12 in total) consumed in the household, including cereals, white roots and tubers, vegetables, fruits, meat and its products, eggs, fish, legumes nuts and seeds, dairy and its products, oils and fats, sweets and sugars, and spices condiments and beverages. The index has been elaborated by Swindale & Bilinsky [50], and recently widely used to study dietary diversity [12, 17, 26, 51]. *Energy, iron, zinc, and vitamin A (micronutrients)* available per adult per household have been measured by computing quantities of food

items consumed by households in kilograms and then computing edible proportions for each food item available. From the edible quantities, we computed quantities of energy in kilocalories and respective micronutrients, following Uganda food consumption tables documented by Hotz et al. [52]. For comparability of nutrition outcomes across households with different demographic compositions, we standardized household size into adult equivalents (AE) following (FAO,WHO & UNU [53], using an adult male as the person category with the highest nutritional requirements to survive. Therefore using this threshold, other persons' nutritional requirements based on their sex (female or male) and age, are computed, and then standardized to their adult male equivalent. Edible quantities of energy and micronutrients were then divided by respective adult equivalents to produce comparable nutrition indicators available per adult across households. Following FAO,WHO & UNU [53], we also computed deficiencies for these micronutrients using as critical levels; 2400 kilocalories, 18 milligrams, 15 milligrams, and 625 retinal activity equivalent micrograms, for energy, iron, zinc, and vitamin A respectively.

**Data description.** From Table 1, the sample was on average aged 48 years, with a household size of 6 persons and was barely educated (1 year of formal education). This being a purely rural sample, low education levels may not be a surprise. The value of annual household assets averaged at seven (7) million UGX (2,000 USD). However, land size averaged at 0.7 acres. Concerning discrete variables, most of the sample (76%) had experienced shocks (weather issues like drought, famine, storms etc., health issues like death of the head, chronical illnesses etc.). Furthermore, most of the sample (63%) used mobile phones, and heavily relied on agriculture (59%) as their main income source. On the other hand, males (66%) dominated household headship. Concerning production diversity, on average, households farmed nearly 5 species of both crops and livestock of which majority (69%) were crop species. However, as seen in Fig 3, the average count of crop species farmed across the years, slightly declined between 2015 and 2019, while that for animals slightly increased over the same period. On the other hand, household dietary diversity score (HDDS) slightly increased from 2015 to 2019, and this variation is illustrated in Fig 4.

Furthermore, average consumption of energy and all considered micronutrients (2,430 kilocalories, 20 milligrams, 13 milligrams, and 701 rae-micrograms respectively) was slightly above FAO recommended thresholds per adult, except for zinc. Energy, iron, and zinc were mostly (64%, 56%, and 66% respectively) sourced from markets, whereas vitamin A was mostly (59%) sourced from own farm produce. From Fig 3, FPD was dominated by crop species perhaps because our sample is totally rural and dominantly composed of smallholder farmers who mostly grow on crops [7, 12, 21, 20, 32]. Overall, the 9 most grown crops species were beans (16.3%), cassava (16.1%), maize (14.9%), cooking bananas (matooke) (13.9%), coffee (8.5%), sweet potatoes (7.8%), groundnuts (3.6%), sorghum (2.7%) finger millet (2.4%), whereas the 5 most farmed animal species were cattle (30.2%), goats (29.5%), chicken (23.3%), pig (8%), and sheep (5.3%). See S1 Table for more details on the composition of the FPD index. We also note that there was slightly more food produced in 2015 than in 2018 or 2019, a fact that is also collaborated by the official data, for instance indicating cereal (maize, rice, sorghum, and millet) production at 4.1 metric tonnes in 2018, and 4.3 in 2019, but 4.5 in 2015 [54–56]. This is also collaborated by FOASTAT data presented in the supplementary data file, the S2 File. We do not expect these differences by time to affect our analysis since we used panel data models to control for time or year specific effects.

## Empirical model for data analysis

We implemented the specification of the panel regression model in Eqs (1) and (2), to study the nature of association between FPD, the two FPD sub-components and various nutrition

**Table 1. Sample descriptive statistics (means or percentages) (N = 6,992).**

| Variables | 2015 (N = 2,381) | | 2018 (N = 2,348) | | 2019 (N = 2,263) | | All sample (N = 6,992) | |
|---|---|---|---|---|---|---|---|---|
| Male head (dummy) | 0.655 | | 0.662 | | 0.669 | | 0.662 | |
| Age of head (years) | 47.91 | (16.14) | 47.69 | (15.90) | 47.78 | (15.94) | 47.80 | (15.99) |
| Household size (persons) | 5.746 | (3.240) | 5.662 | (3.118) | 5.785 | (3.189) | 5.730 | (3.183) |
| Education of head (years) | 1.440 | (3.387) | 1.333 | (3.396) | 1.309 | (3.263) | 1.362 | (3.351) |
| Mobile phone use (dummy) | 0.631 | | 0.628 | | 0.627 | | 0.629 | |
| Total assets (million UGX) | 6.35 | (2.09) | 6.93 | (2.79) | 6.64 | (2.73) | 6.64 | (2.53) |
| Experienced shocks (dummy) | 0.733 | | 0.778 | | 0.783 | | 0.764 | |
| Land Size (Acres by GPS) | 0.686 | (3.609) | 0.682 | (1.858) | 0.624 | (1.742) | 0.664 | (2.565) |
| Farming (main income source) | 0.585 | | 0.585 | | 0.595 | | 0.589 | |
| FPD (bio index) | 4.911 | (3.319) | 4.790 | (3.282) | 4.931 | (3.282) | 4.877 | (3.295) |
| Crops FPD (bio index) | 3.483 | (2.761) | 3.345 | (2.748) | 3.431 | (2.777) | 3.420 | (2.762) |
| Animals' FPD (bio index) | 1.429 | (1.307) | 1.445 | (1.338) | 1.499 | (1.340) | 1.457 | (1.328) |
| HDDS (food groups) | 9.972 | (2.594) | 10.09 | (2.558) | 10.18 | (2.371) | 10.08 | (2.513) |
| Energy (kilocalories/AE) | 3,286 | (2,838) | 2,048 | (1,643) | 1,925 | (2,139) | 2,430 | (2,347) |
| Iron (milligrams/AE) | 25.75 | (23.55) | 17.65 | (13.37) | 17.60 | (18.76) | 20.39 | (19.43) |
| Zinc (milligrams/AE) | 17.36 | (16.70) | 11.20 | (9.260) | 11.42 | (10.33) | 13.37 | (12.90) |
| Vitamin A (rae_micrograms/AE) | 850.3 | (1,217) | 549.6 | (789.0) | 712.2 | (3,504) | 704.6 | (2,168) |
| *From markets* | | | | | | | | |
| Energy (kilocalories/AE) | 2,154 | (2,291) | 1,293 | (1,396) | 1,209 | (1,280) | 1,559 | (1,776) |
| Iron (milligrams/AE) | 14.42 | (17.97) | 9.760 | (10.41) | 10.22 | (10.69) | 11.50 | (13.70) |
| Zinc (milligrams/AE) | 11.53 | (13.75) | 7.203 | (7.863) | 7.747 | (7.976) | 8.851 | (10.46) |
| Vitamin A (rae_micrograms/AE) | 378.7 | (920.5) | 223.7 | (507.8) | 272.7 | (705.9) | 292.4 | (735.2) |
| *From own production* | | | | | | | | |
| Energy (kilocalories/AE) | 1,132 | (1,369) | 755.1 | (879.8) | 716.2 | (1,741) | 870.9 | (1,384) |
| Iron (milligrams/AE) | 11.33 | (13.88) | 7.887 | (9.030) | 7.382 | (15.83) | 8.895 | (13.31) |
| Zinc (milligrams/AE) | 5.834 | (7.503) | 3.994 | (4.931) | 3.676 | (6.555) | 4.518 | (6.492) |
| Vitamin A (rae_micrograms/AE) | 471.6 | (827.4) | 325.9 | (592.4) | 439.4 | (3,440) | 412.3 | (2,045) |
| *Deficiencies* | | | | | | | | |
| Energy | 0.457 | | 0.664 | | 0.699 | | 0.605 | |
| Iron | 0.451 | | 0.602 | | 0.622 | | 0.557 | |
| Zinc | 0.567 | | 0.751 | | 0.722 | | 0.679 | |
| Vitamin A | 0.572 | | 0.727 | | 0.696 | | 0.664 | |

FPD is farm production diversity, HDDS is household dietary diversity score, UGX is Uganda shillings (1USD = 3,557 UGX over considered years), in parentheses are standard deviations. Values without standard deviations are percentages, GPS is global positioning system, AE is adult equivalent, rae_mg is retinal activity equivalents micrograms

outcomes or indicators respectively.

$$HNO_{it} = \alpha_0 + \beta_1 FPD_{it} + \theta X_{it} + \gamma T_t + \varepsilon_{it} \qquad (1)$$

Where $\beta_1$ in this case is the effects of FPD on the nutrition outcome that we estimated.

$$HNO_{it} = \alpha_0 + \beta_1 Animal\_SpeciesCount_{it} + \beta_2 Crop\_SpeciesCount_{it} + \theta X_{it} + \gamma T_t + \varepsilon_{it} \qquad (2)$$

Where $HNO_{it}$ is household nutrition outcome of interest (dietary diversity, energy, zinc, iron, or vitamin A available per adult) of household *i* in year *t*. $\alpha_0$ is the constant. $\beta_1$ and $\beta_2$ are respectively the effects of the animal, and crop species components of FPD that we aim to establish. $\theta$ is a vector of coefficients for observed household, and contextual characteristics,

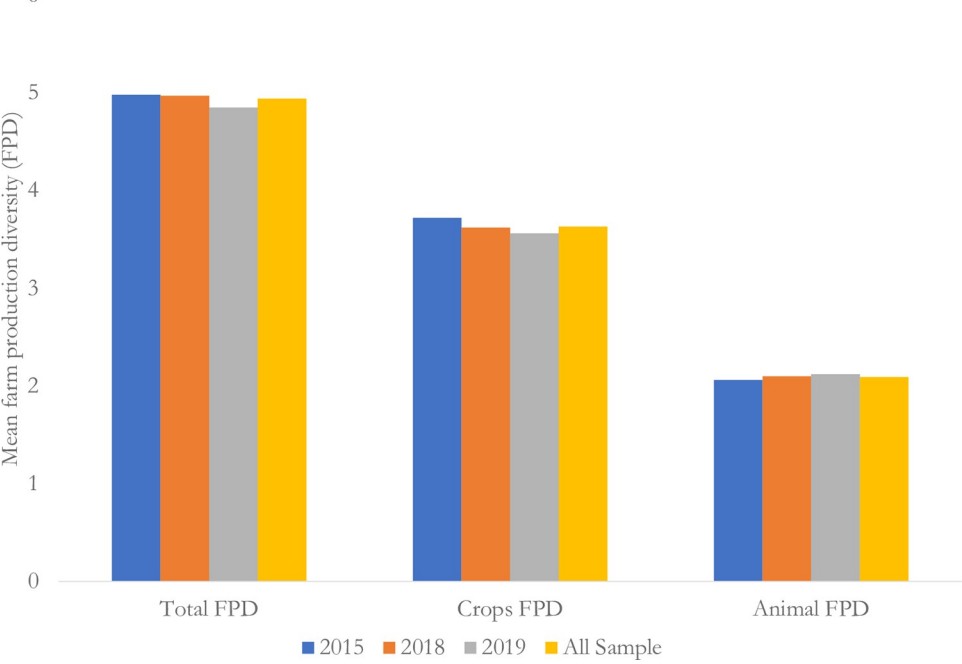

**Fig 3. Farm production diversity (FPD) as generated from different sources (crops or livestock).**

while $\gamma$ is a time fixed effects parameter. $\varepsilon_{it}$ is the normally distributed error term, and $X_{it}$ is the vector of observed household characteristics (education, gender, and age of head, household size, assets, use of mobile phones, major source of income, and exposure to shocks, and farm (land size). These characteristics could alongside the considered FPD or FPD components, influence household nutrition outcomes. $t$ is the year identifier variable capturing yearly fixed effects. We use Eq (2) to empirically study the associations elaborated above for which we do not claim causality.

Although in Eq (1) we controlled for FPD itself, in Eq (2), we controlled for the two sub-components of FPD to examine magnitudes of their coefficients to see which FPD sub-component is associated with better nutrition gains for households. We estimated both Eqs (1) and

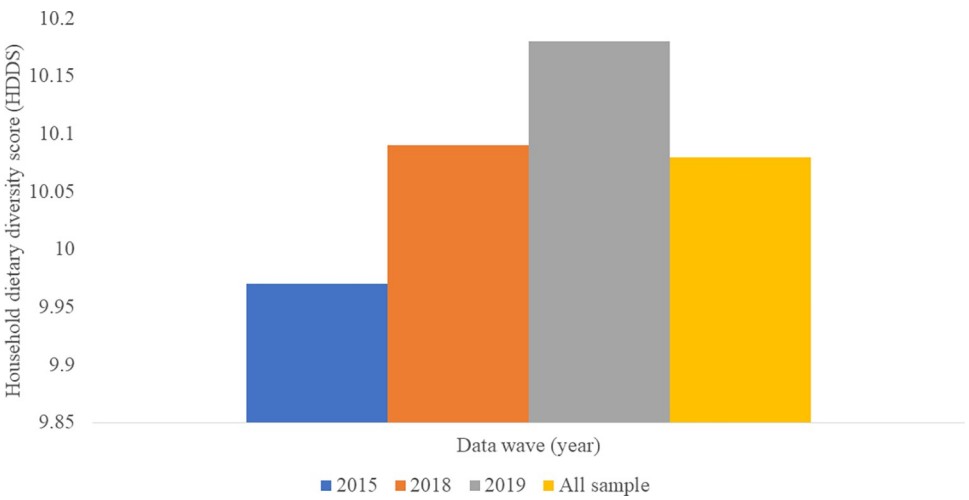

**Fig 4. Variation of household dietary diversity score (HDDS) across the panel waves.**

(2) with random effects (RE) to control for heterogeneity within observed time variant and time invariant household characteristics, and fixed effects (FE) to control for unobserved heterogeneity [57, 58]. Moreover, because the UNPS data is collected randomly, and are a panel, this also helped to reduce potential biases. But, because farmers self-select which crops and livestock species to farm based on own characteristics, and supposedly time-invariant covariates like gender of household heads become variant when headship changes for example due to death or divorce, this may yield systemic bias in results generated by the FE estimator [57]. Moreover, even the RE estimator's strong assumption that FPD cannot correlate with unobserved factors that may influence HDDS, energy, or micronutrients intake is also violated by self-selection [57, 58]. Therefore, to control for potential violations of these assumptions, we use the Mundlak (MK) estimator, a pseudo-fixed effects estimator, that also controls biases caused by time-invariant unobserved heterogeneity, as would do a FE estimator, [59]. Essentially, the MK estimator helps bridge the FE and RE estimations by controlling for means of variables, such that the FE assumption (there is a correlation between specific effects of studied individuals and the independent variables), and the RE assumption (there is no correlation between unobserved heterogeneity of studied individuals and the independent variables) are not violated–which if violated would yield biased estimates [57, 59]. Therefore, we interpret the MK estimator results. Nevertheless, in the first regression results involving HDDS, we present results from both FE and MK estimators for comparisons, and to minimize space we leave out RE results but include the Hausman test value for all FE specifications that consistently showed that the FE model was better suited to the data than the RE specification. However, we only present the MK estimator results concerning energy and micronutrients, to avoid bulkiness. To correctly interpret the MK model, we generate elasticities following Bellemare & Wichman [60], after inverse hypergolic sine (IHS) transformations of the FPD indices. The IHS transformations are preferred for the explanatory variables of interest because they allow retention of preferred properties of the log transformations while retaining the negative and zero-valued observations in the data instead of merely dropping such observations [61].

Therefore, the stated empirical model was appropriately used to find answers to our research questions, using the panel survey data from Uganda covering a representative sample of nearly 3,000 households consisting of the of 2009/10, 2010/2011, and 2011/2012 waves of the Uganda National Panel Survey (UNPS). The data was collected by the Uganda Bureau of Statistics annually and is freely available from the World Bank's Living Standards Measurement Study–Integrated Surveys on Agriculture (LSMS-ISA) section. Because the data is a panel, we used panel data models specifying fixed effects and random effects to analyze the data. These panel data models enable variation in parameters of the model across studied households, which improves efficiency, hence in the quality of estimated results generated from combining households across the different data waves [57, 58]. However, because both the fixed effects and random effects estimators have assumptions that could easily be violated, we finally estimate the Mundlak which sufficiently connects the fixed and random effects estimations [59]. The Mundlak concept is premised on assumptions of the FE and RE estimators [57, 59]. Answers to the above questions have enhanced the understanding of the linkages between FPD and nutrition. Such an interlinked understanding is indispensably important in designing appropriate food systems interventions [16, 29–31]. Since we were focused on studying farm production diversity, we only analyzed rural households of the UNPS. Our results (that crop species count is strongly associated with better nutrition outcomes via the two main consumption pathways–own farm production, and markets) have also generated evidence to inform pro-nutrition and food security policies in Uganda, and those of a similar context, on how inclusiveness in nutrition gains especially among poor rural smallholder farmers can be achieved. More specifically, for instance, our results have among others informed the nutrition

following policy initiatives: 1) The Uganda Nutrition Action Plan II (UNAP II) that spans between 2020–2025, and aims to leave behind none among Ugandans in scaling up nutrition outcomes. 2) The Uganda Food and Nutrition Policy (UFNP). 3) The Uganda National Agriculture Policy (UNAP). 4) Uganda's Multisectoral Food Security and Nutrition Project (UMFSNP) funded by the World Bank aiming mostly to eradicating malnutrition in children and rural dwellers.

## Results and discussions

From Table 2, model 1 we see that the combined (animal and livestock species count) FPD index is significantly and positively associated with HDDS. However, the Hausman test value is significant and for the reasons explained earlier we interpret model 3 the MK specification, which shows that diversification is negatively associated with HDDS. However, this blanket interpretation could be misleading since FPD contains 2 major subcomponents that may influence HDDS differently given the rural context of our sample. We thus re-run the FE and MK estimators each with the two main sub-components of the FPD index to further investigate if each of the two FPD sub-components will differentially be associated with HDDS. On disaggregating the FPD index (crop and livestock) in models 4 and 5 (without controls), and models 6 and 7 (with controls), the two sub-components are associated significantly with HDDS differently. Moreover, in all these models the animal species count shows a positive and significant association (models 4 and 6), while the crop species count shows a negative and significant association. The negative association with crop species could be linked to dominance of crop-farming by most rural households which is also less capital intensive but nearly sufficiently done and usually provides the key needed primary energy sources for households to survive, that more diversification of the same may yield diminishing returns. For quantitative empirical elaborations, we interpret the elasticities in Table 3 for MK estimators particularly model 7.

From model 7, Table 3, each additional animal species grown on farm is associated with increases of 2.4 percentage points in food groups consumed on farm, implying and increment of 0.3 food groups. On the other hand, the association of HDDS with crop species count is negative but with decreases of only 0.8 percentage points on food groups consumed, an implication of 0.1 food groups. Thus, the two sub-components of FPD are differently associated with HDDS. Since our rural sample largely and abundantly grew staple food crops (see S1 Table for details) that are largely cereals or roots and tubers thus contributing mainly to household energy and general micronutrients needs [26, 33–36], it is surprising that a crops species count was significantly and negatively associated with HDDS. However, since largely grown crops species were yielding common dietary quality, for instance the dominant cereals and starchy crops, it may also imply that increments in their species, only yielded diminishing retards on a dietary quality (diversity of diets) indicator, HDDS [17, 18, 62]. On the other hand, animal species count showing a positive and significant association with HDDS is not surprising. Muthini et al. [26], found that producing animals enabled households' access to a diversity of nutrition benefits in energy, proteins, fats, and micronutrients. However, since HDDS is an aggregated indicator of dietary quality [18, 19, 62], we re-run model 7 with specific nutrition outcomes to establish the exact nature of association between FPD sub-components and certain micronutrients. As expected, these unearthed positive and significant associations of both the animal species count and the crop species count with various micronutrients and energy especially those sourced from own-farm produce, that was difficult to detect under a qualitative indicator, HDDS. We present only the elasticities of these results later in respective Tables 4–7, and details are presented in respective S1–S6 Tables.

**Table 2. Association of farm production diversity (FPD) on dietary diversity score (HDDS).**

| Models | RE (1) | FE (2) | MK (3) | FE (4) | MK (5) | FE (6) | MK (7) |
|---|---|---|---|---|---|---|---|
| Variables | HDDS | HDDS | HDDS | HDDS | HDDS | HDDS | HDDS |
| IHS of FPD (bio index) | 0.183*** | -0.210*** | -0.206*** | | | | |
| | (0.032) | (0.062) | (0.056) | | | | |
| IHS of Animal FPD (bio index) | | | | 0.484*** | 0.483*** | 0.485*** | 0.259*** |
| | | | | (0.078) | (0.071) | (0.079) | (0.043) |
| IHS of Crop FPD (bio index) | | | | -0.561*** | -0.557*** | -0.643*** | -0.070** |
| | | | | (0.064) | (0.059) | (0.066) | (0.034) |
| Male head (dummy) | | | | | | 0.030 | -0.004 |
| | | | | | | (0.262) | (0.226) |
| Mobile phone use (dummy) | | | | | | 0.002 | -0.007 |
| | | | | | | (0.107) | (0.093) |
| Age of head (years) | | | | | | -0.021* | -0.007 |
| | | | | | | (0.013) | (0.011) |
| Household size (adult equivalents) | | | | | | 0.013 | -0.005 |
| | | | | | | (0.039) | (0.032) |
| Education of head (years) | | | | | | -0.039 | -0.054** |
| | | | | | | (0.027) | (0.022) |
| Total assets (million UGX) | | | | | | -0.038 | -0.125*** |
| | | | | | | (0.036) | (0.027) |
| Experienced shocks (dummy) | | | | | | 0.104 | 0.075 |
| | | | | | | (0.102) | (0.088) |
| Land Size (Acres by GPS) | | | | | | -0.009 | -0.025 |
| | | | | | | (0.025) | (0.022) |
| Farming is the main income source (dummy) | | | | | | -0.003 | -0.083 |
| | | | | | | (0.111) | (0.096) |
| Year is 2018 | 0.138** | -0.085 | 0.127* | -0.084 | 0.127* | -0.085 | 0.102 |
| | (0.070) | (0.082) | (0.070) | (0.081) | (0.069) | (0.084) | (0.070) |
| Year is 2019 | 0.196*** | -0.107 | 0.186*** | -0.113 | 0.180** | -0.092 | 0.077 |
| | (0.071) | (0.083) | (0.071) | (0.082) | (0.070) | (0.083) | (0.070) |
| *Means of covariates* | | | YES | | YES | | YES |
| Constant | 7.028*** | 7.986*** | 6.672*** | 8.007*** | 6.751*** | 9.220*** | 5.218*** |
| | (0.081) | (0.136) | (0.091) | (0.123) | (0.082) | (0.639) | (0.219) |
| Observations | 6,992 | 6,992 | 6,992 | 6,992 | 6,992 | 6,828 | 6,828 |
| No. of households | 2,838 | 2,838 | 2,838 | 2,838 | 2,838 | 2,804 | 2,804 |
| F value | | 4.46*** | | 23.22*** | | 8.72*** | |
| Hausman test value | | 105.61*** | | 155.98*** | | 158.75*** | |
| Wald Chi2 value | 41.28*** | | 112.39*** | | 217.64*** | | 329.68*** |

Standard errors in parentheses

*** p<0.01

** p<0.05

* p<0.1

UGX is Uganda shillings (1USD = 3,557 USD); GPS is Global positioning system; RE is Random effects, FE is Fixed effects, MK is Mundlak, IHS is Inverse hyperbolic sine. Full table with means of variables is presented in S2 Table.

However, Table 2, model 7 results do also highlight other factors that are significantly associated with HDDS. For instance, our rural sample was barely educated with an average of one (1) year of formal education. Moreover, each additional year of education for such a grossly

**Table 3. Elasticities for the associations of farm production diversity (FPD) on dietary diversity score (HDDS).**

| Nutrition indicator | Household dietary diversity score (HDDS) | | |
|---|---|---|---|
| Models | MK (3) | MK (5) | MK (7) |
| IHS of FPD (bio index) | -0.025*** | | |
|  | (0.006) | | |
| IHS of animals' FPD (bio index) | | 0.045*** | 0.024*** |
|  | | (0.007) | (0.004) |
| IHS of crops FPD (bio index) | | -0.063*** | -0.008** |
|  | | (0.007) | (0.004) |
| *Other model attributes* | | | |
| Other covariates are included | NO | NO | YES |
| Means of covariates | YES | YES | YES |
| Observations | 6,992 | 6,992 | 6,828 |
| No. of households | 2,838 | 2,838 | 2,804 |
| Wald Chi2 value | 112.39*** | 217.64*** | 329.68*** |

Standard errors in parentheses

*** p<0.01

** p<0.05

* p<0.1

IHS is Inverse hyperbolic sine

uneducated sample was negatively associated with reductions in food groups consumed. In fact, all our sample was rural (100%) which is characteristic of strong traditions that are heavily aligned towards consumption of staples and limited education [63, 64]. Therefore, it may not be surprising that associated effects of education towards HDDS were negative–contrary to

**Table 4. Elasticities of associations of farm production diversity (FPD) and daily energy intake per adult equivalent (AE).**

| Nutrition indicator | Daily energy intake (kilocalories/AE) | | | |
|---|---|---|---|---|
| Models | MK (1) | MK (2) | MK (3) | MK (4) |
| Variables | Total | Total | Own farm-sourced | Markets source |
| IHS of FPD (bio index) | 0.034*** | | | |
|  | (0.004) | | | |
| IHS of animals' FPD (bio index) | | -0.003 | 0.031*** | -0.003 |
|  | | (0.004) | (0.009) | (0.006) |
| IHS of crops FPD (bio index) | | 0.039*** | 0.165*** | 0.024*** |
|  | | (0.004) | (0.009) | (0.006) |
| *Other model attributes* | | | | |
| Other covariates | YES | YES | YES | YES |
| Means of covariates | YES | YES | YES | YES |
| Observations | 6,828 | 6,828 | 6,828 | 6,828 |
| No. of households | 2,804 | 2,804 | 2,804 | 2,804 |
| Wald Chi2 value | 986.33*** | 1024.91*** | 1409.65*** | 883.99*** |

Standard errors in parentheses

*** p<0.01

** p<0.05

* p<0.1

MK is Mundlak, IHS is Inverse hyperbolic sine. Full model table is in S3 Table.

our expectations. However, education in substantially more years (higher education of dominantly rural samples), has been found to positively associate with nutrition outcomes [28, 51]. On the other hand, further surprisingly, assets also showed a significant negative association with HDDS. Usually, most assets among smallholder households including productive assets (communication and transport equipment like mobile phones, motorcycles, or bicycles etc.) and non-productive ones are controlled by males who may work largely in non-farm activities [51, 65, 66]. However, considerable household financial resources, that would be used to purchase food or invest in food production–are diverted daily to service costs related to the use of these assets for instance, buying airtime and fuel, repairs etc. Hence, the variable costs burden presented to the household by availability of these assets may render assets to be negatively associated to HDDS. Moreover, even when such assets are liquidated by households, generated incomes are turned to strategic household investments like housing, off-farm business investments, and medication, but not food consumption [51]. Nevertheless, some evidence has found assets to contribute importantly to household welfare [66].

Regarding micronutrients and energy, results are presented in separate tables for each nutrition indicator. From Tables 4–7 model (1), as was expected, FPD in general is positively and significantly associated with total daily energy, iron, zinc, and vitamin A intake per adult equivalent at 1% level. Each additional species of crops and livestock grown on farm increases energy, iron, zinc, and vitamin A available to the household by 3.4 (83 kilocalories per AE), 4.9 (0.99 milligrams/AE), 6.0 (0.8 milligrams/AE), and 2.9 (20.4 rae_micrograms/AE) respectively.4.9 However, in model (2) of each of Tables 4–7, disaggregation of FPD into subcomponents (animal and crop species), shows differential associations with micronutrients and energy, some of which is insignificant. Nevertheless, the insignificance of the associations between micronutrients and some FPD subcomponents could be due to aggregation of micronutrients irrespective of the source (own farm and markets), yet we consider only rural households that mostly rely on one source (own farm produce) for their nutrition needs. Therefore, in models (3) and (4) of Tables 4–7, we analyze FPD subcomponents and their associations with energy and micronutrients by source.

More specifically, from Table 4, when FPD was disaggregated into the two subcomponents in model (2) while energy was still aggregated, the association with energy is negative with the animal species count but insignificant but positive and significant with crops species count (3.9 percentage points). However, in models 3 and 4 when total energy is disaggregated by source–own farm and markets source, the animal species count then shows a strong significant and positive association with energy sourced from own farm produce consumption. Specifically, each additional animal species kept on farm is associated with increases in daily energy intake per adult equivalent of 3.1 (27 kilocalories/AE) percentage point. The association of the animal species count with energy sourced from markets is not significant.

Similarly, from Table 4, models (3) and (4) the association of the crops' species count with energy sourced from both own farm produce and markets consumption was strongly significant and positive, with each additional crops' species being associated with an additional 16.5and 2.4 percentage points via each source respectively. Associations with energy sourced from own farm produce yielded starker increments.

From Table 5, disaggregating FPD while iron was aggregated in model 2, the animal species count shows negative but insignificant associations with iron intake. However, the crop species count shows a significant and positive association with increments of 5.3 percentage points in food groups consumed for every additional crop species. When iron is disaggregated by source, again, both the animal and crops species count show a significant and positive association with daily iron intake sourced from own farm. Each additional animal species is associated with an additional 10.9 percentage points, while that of crops is associated with an

**Table 5. Elasticities of associations of farm production diversity (FPD) and daily iron intake per adult equivalent (AE).**

| Nutrition indicator | Daily iron intake (milligrams/AE) | | | |
|---|---|---|---|---|
| Models | MK (1) | MK (2) | MK (3) | MK (4) |
| Variables | Total | Total | Own farm-sourced | Markets source |
| IHS of FPD (bio index) | 0.049*** | | | |
| | (0.007) | | | |
| IHS of animals' FPD (bio index) | | -0.0004 | 0.109** | 0.004 |
| | | (0.007) | (0.043) | (0.013) |
| IHS of crops FPD (bio index) | | 0.053*** | 0.752*** | 0.025** |
| | | (0.007) | (0.042) | (0.012) |
| *Other model attributes* | | | | |
| Other covariates | YES | YES | YES | YES |
| Means of covariates | YES | YES | YES | YES |
| Observations | 6,828 | 6,828 | 6,828 | 6,828 |
| No. of households | 2,804 | 2,804 | 2,804 | 2,804 |
| Wald Chi2 value | 581.57*** | 602.69*** | 1298.19*** | 284.01*** |

Standard errors in parentheses

*** $p < 0.01$

** $p < 0.05$

* $p < 0.1$

MK is Mundlak, IHS is Inverse hyperbolic sine. Full table is in S4 Table.

additional 75.2 percentage points on the daily iron intake sourced from own farm. Associations with iron sourced from markets were insignificant although positive with animal species count, but significant and positive with crop species count, where each additional species was associated with increases 2.5 percentage points in iron intake sourced from markets.

**Table 6. Elasticities of associations of farm production diversity (FPD) and daily zinc intake per adult equivalent (AE).**

| Nutrition indicator | Daily zinc intake (milligrams/AE) | | | |
|---|---|---|---|---|
| Models | MK (1) | MK (2) | MK (3) | MK (4) |
| Variables | Total | Total | Own farm-sourced | Markets source |
| IHS of FPD (bio index) | 0.060*** | | | |
| | (0.009) | | | |
| IHS of animals' FPD (bio index) | | 0.002 | 0.753*** | 0.005 |
| | | (0.009) | (0.222) | (0.015) |
| IHS of crops FPD (bio index) | | 0.065*** | 0.812*** | 0.050*** |
| | | (0.009) | (0.216) | (0.015) |
| *Other model attributes* | | | | |
| Other covariates | YES | YES | YES | YES |
| Means of covariates | YES | YES | YES | YES |
| Observations | 6,828 | 6,828 | 6,828 | 6,828 |
| No. of households | 2,804 | 2,804 | 2,804 | 2,804 |
| Wald Chi2 value | 529.53*** | 548.02*** | 1353.46*** | 396.52*** |

Standard errors in parentheses

*** $p < 0.01$

** $p < 0.05$

* $p < 0.1$

MK is Mundlak; IHS is Inverse hyperbolic sine. Full table is in S5 Table.

**Table 7. Elasticities of associations of farm production diversity (FPD) and daily vitamin-A intake per adult equivalent (AE).**

| Nutrition indicator | Daily vitamin-A intake (rae_micrograms/AE) | | | |
|---|---|---|---|---|
| Models | MK (1) | MK (2) | MK (3) | MK (4) |
| Variables | Total | Total | Own farm-sourced | Markets source |
| IHS of FPD (bio index) | 0.029*** | | | |
| | (0.006) | | | |
| IHS of animals' FPD (bio index) | | 0.021*** | 0.051*** | 0.038*** |
| | | (0.006) | (0.017) | (0.012) |
| IHS of crops FPD (bio index) | | 0.019*** | 0.260*** | -0.011 |
| | | (0.006) | (0.017) | (0.011) |
| *Other model attributes* | | | | |
| Other covariates | YES | YES | YES | YES |
| Means of covariates | YES | YES | YES | YES |
| Observations | 6,828 | 6,828 | 6,828 | 6,828 |
| No. of households | 2,804 | 2,804 | 2,804 | 2,804 |
| Wald Chi2 value | 240.75*** | 247.29*** | 1062.00*** | 212.63*** |

Standard errors in parentheses

*** $p < 0.01$

** $p < 0.05$

* $p < 0.1$

MK is Mundlak; IHS is Inverse hyperbolic sine. rae_mg is retinal activity equivalents micrograms. Full table is in S6 Table.

From Table 6, when FPD was disaggregated while zinc was still aggregated, again, the animal species count was insignificantly associated with zinc, although positively. On the other hand, the crop species count is significant and positively associated with daily zinc intake with each additional crop species being associated with increments of 6.5 percentage points in zinc intake. However, when zinc is disaggregated by source, both the animal and crops species count show a strongly significant and positive association with zinc sourced from own farm. Each additional species to animals and crops grown is associated respectively with an additional 75.3 and 81.2 percentage points on daily zinc intake sourced from own farms. The large size of these incremental margins could be explained by the high (highest of all studied nutrition indicators) levels of zinc deficiency in the sample, as well as Ugandan population at large [28, 33, 35]. Associations of animal species count with zinc sourced from markets were insignificant but positive. However, such association was positive and significant for the crop species count, where each additional crop species was associated with increments of 5 percentage points in daily zinc intake.

Taking a slightly different pattern, from Table 7, when FPD was disaggregated while vitamin A was still aggregated, both the animal and crop species count yields a positive and strongly significant association with daily vitamin A intake. Each additional animal and crop species grown yields respectively an associated additional 2.1 and 1.9 percentage points of total daily vitamin A intake per adult. Furthermore, if total daily vitamin intake is disaggregated by source, like with other micronutrients and energy, both animal and crops species count independently yield a positive and strongly significant association with vitamin A sourced from own farm foods consumption. Each additional species for animals and crops grown is associated respectively with an additional 5.1 and 26 percentage points to daily vitamin intake sourced from own farm. Unlike with earlier nutrition indicators studied here, the associated effect of the animal species count with vitamin A sourced from markets is positive and strongly significant. Each additional animal species on the farm, is associated with 3.8 percentage points

increments in daily vitamin A intake sourced from markets. The association with crops species count was negative but insignificant, unlike other micronutrients.

Generally, from Tables 4–7, aggregated FPD showed significant and positive associations with total daily energy, iron, zinc, and vitamin A intake. When FPD was disaggregated in the two components–the animal species count, showed insignificant associations with total energy and totals of other micronutrients except vitamin A. At this level, all associations between the crop species count and the aggregated totals of energy and all micronutrients were all positive and strongly significant. However, when total energy and totals of micronutrients were also disaggregated by source (own farm or markets), and individually analyzed alongside disaggregated FPD, both the animal and the crops species count, consistently exhibited a positive and strongly significant association with household daily energy and micronutrients intake sourced from own farm consumption. In all cases, incremental percentage points yielded by each additional crop species were larger than those yielded by each additional animal species grown on farm. Moreover, FPD in general has been previously found to be positively associated with nutrition outcomes [8, 15, 17, 24–29, 62]. In disaggregated terms, small animals kept on farm like goats, sheep, and rabbits and poultry species like chicken and ducks that formed most of the animal species count can easily be consumed for food within households anytime of the year without their availability being dependent on farming seasons. Moreover, larger animals like cattle and even small animals can regularly provide products like milk, and eggs that are good sources of energy and micronutrients. Therefore, regular consumption of animals and their products, makes it possible for households to enhance their available energy and micronutrients. Our findings agree with Muthini et al. [26] who found the animal species count to be important to household dietary diversity. Gaillard et al. [8] also found that women consumed more dairy products if these were produced on farm. However, although the association is as expected negative for energy, and positive for iron and zinc, the animal species count is not significantly associated with daily energy, iron, and zinc intake sourced from markets. Surprisingly, the association was positive and strongly significant with vitamin A sourced from markets.

On the other hand, the crop species count showed a consistent, strongly significant, and positive association with daily energy and micronutrients intake regardless of the source–except for vitamin A sourced from markets. The association was strongly significant with energy and all micronutrients sourced from own farm consumption. In fact, the associations of FPD components (animal and crop species count) and nutrition indicators, were strongest via those components of nutrition indicators that had been sourced from own farm consumption. Considering the two FPD components, the crop species count showed the strongest associations with nutrition indicators within a particular source (except for vitamin A sourced from markets) and across sources (own farm vs. markets). For example, each additional crop species count yielded 16.5 incremental percentage points of daily energy intake via own farm sources compared to 2.4 via markets. The strong positive association of the crop species count with energy and micronutrients sourced from own farm produce is not surprising since most smallholder farmers are engaged in subsistence agriculture, and our sample was totally rural. Hence one would expect that since our sample mostly consume what they grow, then a crop species count should bear a strong positive association with nutrition outcomes, as has been established previously [11, 12, 15, 17–19, 24, 26, 28]. Moreover, the crop species count has also showed positive and strongly significant associations with energy, iron, and zinc intake sourced from markets, despite this being a dominantly rural sample with poor market infrastructure Such consistent, positive and significant associations, may further confirm the importance of the markets consumption pathway which is only possible to farmers after gaining income from selling their produce, in this case crops as has been asserted in literature [12,

17–19, 24, 28]. In good seasons, farmers sell their surplus crops or sell cash crops in all seasons to accumulate income that is used in purchasing foods from markets [33–36]. Moreover, some strategically valuable crops like coffee, and vegetables are farmed within households but sold to earn money and smooth other consumption and non-consumption needs [25, 36, 67, 68]. Yet, households usually never regularly consume such valuable crops like vegetables within households but spare them for sale, and usually consider it a luxury to consume these crops [33, 34, 36]. Such may also explain why associations of the crop species and vitamin A derived from market sources was negative and insignificant. Our findings partly concur with two important meta-analysis reviews around farm production diversity and food consumption–that indeed there are strong (significant) associations between farm production diversity and certain nutrition outcomes as Jones [29] established, which are mostly realized via the diversification of crops species contributing towards own farm sources of energy and micronutrients intake. However, the association is also positive and significant, but the proportional magnitudes of these associations are small, as found Sibhatu and Qaim [18], especially through the animal species count component of the FPD via the own farm and market sources, and the crop species count component contributing to energy and micronutrients intake via the market sources.

## Policy implications

We cautiously point out to policy that investments (scientific, technical, physical, and financial) in farmers' own farm production of both livestock and crop species in rural Uganda, do positively influence nutrition outcomes of households. Therefore, wherever such investment opportunities are available–whether from national, regional, or international governments, then these must be harnessed and carefully scaled-out. However, for the contextual structure of rural Uganda (remote with poor roads and poor market infrastructure), investments in production of crops species at farm household level yield better nutritional outcomes than investments in livestock species–especially via the own farm produce consumption pathway. Therefore, for the government of Uganda, and those partners who are flexible with what they support, the emphasis could be put on diversification in production of crop species. However, our analysis did not consider what crop or animal species were more feasible than others regarding the studied nutrition outcomes, hence we are cautious not to make specific crops or animal species' recommendations. Moreover, this can be an interesting area of research to be exploited in the future. Nevertheless, on-going food security and nutrition initiatives can be effectively guided by this research on the general areas of prioritization for instance, prioritizing investments in crops species diversification. Strategic examples of these policy initiatives among others include: 1) The Uganda Nutrition Action Plan II (UNAP II) that spans between 2020–2025, and aims to leave behind none among Ugandans in scaling up nutrition outcomes. 2) The Uganda Food and Nutrition Policy (UFNP). 3) The Uganda National Agriculture Policy (UNAP). 4) and the Uganda's Multisectoral Food Security and Nutrition Project (UMFSNP).

## Study limitations

From literature some of which was cited in this paper; indeed, the population consumes fewer protein foods as would be expected of rural populations, and we agree that studying protein consumption patterns would be noble. Unfortunately, we did not study proteins for this study. This is because, we build on the work of Sekabira et al. [28], where in the original data, did not compute for proteins. However, we make a stark recommendation to expand consideration of these strategic macro and micronutrients in empirical studies on nutrition. Even though such

wider coverage of many macro and micronutrients may be difficult in one paper, but done individually for each macro or micronutrient would suffice, to avoid getting lost into so much detail.

## Conclusions

Using nationally representative rural households' panel data from Uganda, we establish that indeed, FPD in general is positively and significantly associated with HDDS. However, the two sub-components of FPD (animal species count and crops species count) are differentially associated with HDDS. For instance, the animal species count is significantly and positively associated with HDDS, while the crop species count showed a negative association. Differential and insignificant associations could stem from the high categorization (aggregation) embedded in HDDS. On analyzing nutrition indicators (energy, iron, zinc, and vitamin A) embedded in HDDS, singly, and disaggregating these by source (own farm vs. markets), the crops species count is significantly and positively associated with all studied micronutrients and energy irrespective of their source–except for vitamin A sourced from markets where the association is negative but insignificant. On the other hand, the animal species count is also significantly and positively associated with energy and all micronutrients sourced from consumption of own farm produce, and vitamin A sourced from markets. However, such association is insignificant but positive for zinc and iron sourced from markets, while it is negative for energy. The wider significance of the crops' species count clearly highlights the strategic importance of crops towards better smallholder households' nutrition–especially in a rural sample like ours, where remoteness and poor market infrastructure are persistently prevalent. Crops can easily be consumed directly or sold to markets for income to buy other food items. Generally, concerning individual micronutrients and energy intake, the crop species count shows a stronger association via the own farm produce consumption pathway. Therefore, in a smallholder farmer context, diversification in crop species could be more important than animal species diversification towards availing more energy, and micronutrients per adult. Hence, comparative efforts (household or policy level) targeted towards crop species diversification in farm production could still yield better household nutrition outcomes. However, notice should be taken that our sample is fully rural, and traditionally dominantly reliant on crops than animals to satisfy their food needs and general livelihoods. Hence, our results may not be binding in a context of countries that are predominantly dependent on animals (pastoralists) hence, must be interpreted cautiously in dominantly pastoral and urban contexts.

## Supporting information

**S1 Table. Animal and crops species used in the calculation of the FPD bio index.**
(DOCX)

**S2 Table. Association of farm production diversity (FPD) on Household dietary diversity score (HDDS).**
(DOCX)

**S3 Table. Association of farm production diversity (FPD) and daily energy intake per adult equivalent (AE).**
(DOCX)

**S4 Table. Association of farm production diversity (FPD) and daily iron intake per adult equivalent (AE).**
(DOCX)

**S5 Table. Association of farm production diversity (FPD) and daily zinc intake per adult equivalent (AE).**
(DOCX)

**S6 Table. Association of farm production diversity (FPD) and daily vitamin-A intake per adult equivalent (AE).**
(DOCX)

**S1 File. Data used for the generation of these model results.**
(DTA)

**S2 File. Data from FAOSTAT used to further show that food production in 2015 was generally more than that in 2018 or 2019.**
(CSV)

## Acknowledgments

We are grateful to the World Bank for making this data available, and UBOS for collecting the data. We also gratefully acknowledge Heath Henderson, and our anonymous reviewers for their immensely constructive comments.

## Author Contributions

**Conceptualization:** Haruna Sekabira.

**Data curation:** Haruna Sekabira, Zainab Nansubuga.

**Formal analysis:** Haruna Sekabira.

**Funding acquisition:** Haruna Sekabira.

**Investigation:** Haruna Sekabira.

**Methodology:** Haruna Sekabira.

**Project administration:** Haruna Sekabira, Zainab Nansubuga, Stanley Peter Ddungu, Lydia Nazziwa.

**Resources:** Haruna Sekabira, Zainab Nansubuga.

**Software:** Haruna Sekabira.

**Supervision:** Haruna Sekabira, Zainab Nansubuga, Stanley Peter Ddungu.

**Validation:** Haruna Sekabira, Stanley Peter Ddungu, Lydia Nazziwa.

**Visualization:** Haruna Sekabira, Zainab Nansubuga, Stanley Peter Ddungu, Lydia Nazziwa.

**Writing – original draft:** Haruna Sekabira.

**Writing – review & editing:** Haruna Sekabira, Zainab Nansubuga, Stanley Peter Ddungu, Lydia Nazziwa.

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
