## [Decision Letter · Decision Letter 0]

12 Apr 2022

PONE-D-22-06649Farm Production Diversity, Household Dietary Diversity and Nutrition: Evidence from Uganda’s National Panel SurveyPLOS ONE

Dear Dr. Sekabira,

Thank you for submitting your manuscript to PLOS ONE. After careful consideration, we feel that it has merit but does not fully meet PLOS ONE’s publication criteria as it currently stands. Therefore, we invite you to submit a revised version of the manuscript that addresses the points raised during the review process.

We look forward to receiving your revised manuscript.

Kind regards,

Festo Massawe

Academic Editor

PLOS ONE

Journal Requirements:

[This research was financially supported by the 50x2030 Initiative through the International Fund for Agricultural Development (IFAD), under application no: 2104060035. We also gratefully acknowledge Heath Henderson for his constructive review and comments.]

 [HS

2104060035

International Fund for Agricultural Development

https://www.ifad.org/en/

The funders played no role in writing the manuscript]

Reviewers' comments:

Reviewer's Responses to Questions

**Comments to the Author**

1. Is the manuscript technically sound, and do the data support the conclusions?

Reviewer #1: Partly

Reviewer #2: Yes

Reviewer #3: Yes

2. Has the statistical analysis been performed appropriately and rigorously? 

Reviewer #1: Yes

Reviewer #2: I Don't Know

Reviewer #3: Yes

3. Have the authors made all data underlying the findings in their manuscript fully available?

Reviewer #1: Yes

Reviewer #2: Yes

Reviewer #3: Yes

4. Is the manuscript presented in an intelligible fashion and written in standard English?

Reviewer #1: Yes

Reviewer #2: Yes

Reviewer #3: Yes

5. Review Comments to the Author

Reviewer #1: Review of: Farm Production Diversity, Household Dietary Diversity and Nutrition: Evidence from Uganda’s National Panel Survey (Manuscript Number: PONE-D-22-06649)

This paper examines associations between the diversity of crop and livestock products produced by households living in Uganda and measures of the diversity of their food consumption. Although there are already many papers published on this topic, this is a welcome addition to the literature as it includes new evidence from a country that previously had received relatively little attention. That said, the paper would benefit from revisions along a number of lines.

(1) The discussion section needs to be more clear as to how the results presented in this paper fit into the literature. Specifically, I would strongly encourage the authors to place their paper within the debate found in a comparison of the review papers by Jones, A.D., 2017. (Critical review of the emerging research evidence on agricultural biodiversity, diet diversity, and nutritional status in low- and middle-income countries. Nutrition. Reviews. 75, 769–782) and Sibhatu, K. T., & Qaim, M. 2018. (Review: Meta-analysis of the association between production diversity, diets and nutrition in smallholder farm households. Food Policy, 77, 1–18.) Both papers provide empirical reviews on the links between production diversity and diversity of food consumption but reach different conclusions. Jones sees a “strong” link between production and consumption diversity; by contrast, Sibhatu and Qaim argue that “The average marginal effect of production diversity on dietary diversity is positive but small”.

My sense is that the paper’s findings are more consistent with the view put forward by Sibhatu and Qaim. Put differently, the authors do not give sufficient attention to the magnitudes of the associations that they observe in their data, both in terms of contextualizing their findings and also in their description of the policy implications. If indeed the magnitudes are small, then it is not obvious that so much attention should be paid to encouraging production diversification.

(2) It is not entirely clear if the sample includes both urban and rural households. This is not stated in the description of the survey, though a urban dummy variable appears in the reported regression results. My strong suggestion would be to focus solely on rural households. Urban households generally grown little food of their own and my guess is that those who do tend to be less well-off (which may explain the odd results the authors sometimes get on the crop diversity variable).

(3) Regional dummy variables appear in the regression results; it is not clear why they are being reported when the estimator is a household fixed effect specification (which should difference out time invariant characteristics such as location).

(4) The description of the survey suggests that data are collected twice per year. It is not clear how this is accounted for in the model specification.

(5) Does the measure of crop diversity include non-edible crops such as coffee and cotton? Given the focus of this paper, crop diversity should only include edible crops.

(6) Focusing on Table 2, how much does HDDS vary over time for any given household. Particularly with a fixed effect specification, I wonder if one of the reasons why the parameter estimates are so small is because for many households, there is no change in HDDS over time. A similar concern applies to the index variables for FPD, crops and animals.

(7) A strength of the paper is that it pays attention to the potentially confounding role of time invariant effects, specifically through the use of random effects, fixed effects and the Mundlak pseudo-fixed effects estimators. But strikingly, all three produce very similar parameter estimates. Given this, and given that the test statistics consistently report rejecting random effects in favor of a fixed effects specification, I suggest dropping the random effects results from the paper and retaining the results from the other two estimation methods. For Table 2, this change will create some space that can be used to present additional results. I suggest that the authors show the following:

Column 1: FE model with FPD bio index, no other controls

Column 2: MK model with FPD bio index, no other controls

Column 3: FE model with animal bio index & crop bio index, no other controls

Column 4: MK model with animal bio index & crop bio index, no other controls

Column 5: FE model with animal bio index & crop bio index, full set of controls

Column 6: MK model with animal bio index & crop bio index, full set of controls

This set of specifications will allow your readers to see that the results are not sensitive to either the use of FE or MK, or to the inclusion/exclusion of other control variables

(8) I suggest that the authors break Table 3 into four parts (one for each micro-nutrient). For each micro-nutrient, I suggest that they present four results:

Column 1: MK model. Total intake (=Own farm + market) with FPD bio index, full set of controls

Column 2: MK model. Total intake (=Own farm + market) with animal bio index & crop bio index, full set of controls

Column 3: MK model. Own farm intake with animal bio index & crop bio index, full set of controls

Column 4: MK model. Market intake with animal bio index & crop bio index, full set of controls

This reporting would be more informative than what currently exists in the paper. It would allow you to see the total effect of production diversity on micro-nutrient consumption (column 1); how much of this effect comes from animals and how much from crops (column 2); and whether own production is crowding out market intake (by comparing results from columns 3 and 4)

Reviewer #2: This is a relevant manuscript that highlights the importance of producing diverse food products. My comments for consideration are below:

• More information is needed in the introduction section describing the common food production and consumption patterns in Uganda. What various types of crops and livestock are usually produced? This is important because HDD score depends on food groups. Also, what proportion of farmers eat their own farm produce and what proportion do not. In other words, what proportion of own production is consumed? What types of farm products are usually sold for cash and not consumed by farmers?

• A few specific examples of crops considered in the counts would be useful since nutrient content and expected nutritional contributions may differ among crops. Such information will be helpful in the discussions on page 17.

• The authors found a strong positive association of the crop species count with energy and micronutrients sourced from own farm produce and attributed this to farmers mostly consuming what they grow. So, my question is: what is the problem in Uganda? If farmers are producing and eating what they grow, then what is the gap? Why are they hungry or suffering from malnutrition?

• In the last sentence on Page 18, what do you mean by essential foods? You listed cereals or their products, oils and fats etc. Why do you refer to these foods as essential? Also, you need to distinguish between raw produce and processed products. Aren't the farmers producing cereals and foods rich in fats and oils e.g. groundnuts? Clarify this essential food category.

• You concluded that diversification in crop species could be more important than animal species diversification towards availing more energy, and micronutrients per adult. Were you expecting significant differences in nutritional contrition of different livestock species? What different species were counted? Perhaps, livestock and fisheries could have some differences in nutritional value and contributions to HDDS. If there is data on fish production and consumption, then it will be interesting and relevant to consider it in the analysis.

• Other comments: the manuscript needs to be better structured with introduction, methods and data analysis sections clearly separated. Last paragraph on page 4 – 5 could be moved to methods section. Information under your current methods section could be put under data analysis.

Reviewer #3: The manuscript addresses an area of great importance to the success of policies, especially in resource-scarce settings. The manuscript is well-organised and easy to follow. I have read the manuscript and have made the following observations which I think the authors need to address before publication can be considered:

1. The manuscript requires significant proofreading, especially checking of punctuation.

2. Section I: Introduction

a) Please justify why the study only investigated the impact of FPD on daily energy, iron, zinc, and vitamin A, but not any other nutrients. Particularly, how do these relate to the nutritional issues in Uganda?

3. Section C: Data Description

a) It would be helpful for reader to give an overview of farm characteristics of the sample, e.g., location on map, names and types of crops cultivated, cropping system etc.

b) Referring to Table 1, the amount of energy and most micronutrients consumed from both market and own farm sources are far higher in year 2015 than in 2018 and 2019. Is there any explanation to this anomaly? Could this anomaly affect the statistical rigour?

4. Methods, Results and Discussions

a) It appears that the volume of crop yield and its potential confounding effect on HDDS and nutritional outcome have not been considered or discussed sufficiently in the manuscript.

b) “Each crop species added to those farmed with in a household was associated with 5.9 and 18.8 kilocalories (0.6 and 0.9 percentage points) added to energy sourced from own farm produce consumption or markets respectively. With regards to iron, each additional crop species was associated with 0.4 and 0.1 milligrams (4.2 and 0.7 percentage points) added to daily iron intake sourced from own farm produce or markets consumption respectively. With regards to zinc, each additional crop species grown on the farm was associated with 0.3 and 0.1 milligrams (6.2 and 0.9 percentage points) added to daily zinc intake sourced from own farm produce, and markets consumption respectively. Lastly, each additional crop species grown on farm was associated with 1.1 and 0.6 rae_micrograms (0.3 and 0.2 percentage points) added to daily vitamin A intake sourced from own farm produce, and markets consumption respectively.”

What is the significance of the percentages of gains quoted above in the diets of the sample population? Are these considered meaningful impact on nutritional status? Also, will increment in crop diversity continue to produce nutritional gains at a linear rate or will the effect level off? At what point will the effect level off?

c) “There are however other factors that are consistently and significantly associated with energy, and micronutrients intake, for instance gender effects, household size, and year variables, which we don’t discuss here to prioritize our focus on FPD sub-components, which are our main covariates.”

Wouldn’t understanding these other factors that are also consistently and significantly associated with energy, and micronutrients intake and the relationships of these factors with FPD help in devising more effective policy?

5. Section VI: Policy Implications

a) This section is not sufficiently discussed. It is suggested that the government should devote more attention to crop species, but it does not recommend any specific measures. The authors should focus on how to translate their research into practice at varying levels of scale, considering all other confounders and trade-offs.

b) For various apparent reasons, the policymakers might be more inclined to invest in promotion of cash crops as a means to achieve rural development and poverty alleviation. Cash crops also encourages mono-cropping that reduces FPD. How can this dilemma be addressed? How can policy makers be convinced?

6. PLOS authors have the option to publish the peer review history of their article (what does this mean?). If published, this will include your full peer review and any attached files.

Reviewer #1: No

Reviewer #2: No

Reviewer #3: No

---

## [Author Response · Author response to Decision Letter 0]

27 May 2022

Editors:

The title page and the general manuscript have been reshaped based on the templates from PLOS ONE. Funding information has also been clearly indicated in the cover letter.

Reviewer #1: Review of: Farm Production Diversity, Household Dietary Diversity and Nutrition: Evidence from Uganda’s National Panel Survey (Manuscript Number: PONE-D-22-06649). This paper examines associations between the diversity of crop and livestock products produced by households living in Uganda and measures of the diversity of their food consumption. Although there are already many papers published on this topic, this is a welcome addition to the literature as it includes new evidence from a country that previously had received relatively little attention. That said, the paper would benefit from revisions along a number of lines.

(1) The discussion section needs to be more clear as to how the results presented in this paper fit into the literature. Specifically, I would strongly encourage the authors to place their paper within the debate found in a comparison of the review papers by Jones, A.D., 2017. (Critical review of the emerging research evidence on agricultural biodiversity, diet diversity, and nutritional status in low- and middle-income countries. Nutrition. Reviews. 75, 769–782) and Sibhatu, K. T., & Qaim, M. 2018. (Review: Meta-analysis of the association between production diversity, diets and nutrition in smallholder farm households. Food Policy, 77, 1–18.) Both papers provide empirical reviews on the links between production diversity and diversity of food consumption but reach different conclusions. Jones sees a “strong” link between production and consumption diversity; by contrast, Sibhatu and Qaim argue that “The average marginal effect of production diversity on dietary diversity is positive but small”. This comparison is clearly now done but placing our paper well in-between these two papers – that indeed we find strong (significant) associations between farm production diversity and nutrition outcomes, but the proportional magnitudes of these increments are small – and mostly realized from the diversity in crops species.

My sense is that the paper’s findings are more consistent with the view put forward by Sibhatu and Qaim. Put differently, the authors do not give sufficient attention to the magnitudes of the associations that they observe in their data, both in terms of contextualizing their findings and also in their description of the policy implications. If indeed the magnitudes are small, then it is not obvious that so much attention should be paid to encouraging production diversification. Sometimes, small magnitudes can be substantial proportionally and where we achieve these, especially with crop species for instance with iron 3.4% and zinc 6% proportional increments – indeed it becomes plausible to encourage diversification in such sources.

(2) It is not entirely clear if the sample includes both urban and rural households. This is not stated in the description of the survey, though a urban dummy variable appears in the reported regression results. We now clearly state that the sample is rural and we leave out all urban households. My strong suggestion would be to focus solely on rural households. This guidance is effected. Urban households generally grown little food of their own and my guess is that those who do tend to be less well-off (which may explain the odd results the authors sometimes get on the crop diversity variable).

(3) Regional dummy variables appear in the regression results; it is not clear why they are being reported when the estimator is a household fixed effect specification (which should difference out time invariant characteristics such as location). Regional dummies are removed and FE re-estimated.

(4) The description of the survey suggests that data are collected twice per year. It is not clear how this is accounted for in the model specification. We clearly state that data is collected once a year even though the naming of the waves, coincided with the planning years of UBOS, an entity which collects the data

(5) Does the measure of crop diversity include non-edible crops such as coffee and cotton? Given the focus of this paper, crop diversity should only include edible crops. Because we have a market component/consumption pathway, aided by sales of both edible and non-edible crops, we included only coffee – which is also sometimes roasted and eaten in households. Essentially, we include only edible crops.

(6) Focusing on Table 2, how much does HDDS vary over time for any given household. Particularly with a fixed effect specification, I wonder if one of the reasons why the parameter estimates are so small is because for many households, there is no change in HDDS over time. A similar concern applies to the index variables for FPD, crops and animals. We add a figure to show the variation of HDDS – and further yearly values depicting variations are elaborated in Table 1. It is also made clear that data was collected once every year.

(7) A strength of the paper is that it pays attention to the potentially confounding role of time invariant effects, specifically through the use of random effects, fixed effects and the Mundlak pseudo-fixed effects estimators. But strikingly, all three produce very similar parameter estimates. Given this, and given that the test statistics consistently report rejecting random effects in favor of a fixed effects specification, I suggest dropping the random effects results from the paper and retaining the results from the other two estimation methods. This is perfectly considered and has reduced the columns in Table 2, to a sizeable 6. For Table 2, this change will create some space that can be used to present additional results. I suggest that the authors show the following:

Column 1: FE model with FPD bio index, no other controls

Column 2: MK model with FPD bio index, no other controls

Column 3: FE model with animal bio index & crop bio index, no other controls

Column 4: MK model with animal bio index & crop bio index, no other controls

Column 5: FE model with animal bio index & crop bio index, full set of controls

Column 6: MK model with animal bio index & crop bio index, full set of controls

This set of specifications will allow your readers to see that the results are not sensitive to either the use of FE or MK, or to the inclusion/exclusion of other control variables This has been perfectly adapted, and indeed as the reviewers advised this is now well displayed by our results that they are consistent despite inclusions or exclusions of other controls

(8) I suggest that the authors break Table 3 into four parts (one for each micro-nutrient). For each micro-nutrient, I suggest that they present four results:

Column 1: MK model. Total intake (=Own farm + market) with FPD bio index, full set of controls

Column 2: MK model. Total intake (=Own farm + market) with animal bio index & crop bio index, full set of controls

Column 3: MK model. Own farm intake with animal bio index & crop bio index, full set of controls

Column 4: MK model. Market intake with animal bio index & crop bio index, full set of controls

This reporting would be more informative than what currently exists in the paper. This is perfectly considered, and now each micronutrient is presented in own Table. It would allow you to see the total effect of production diversity on micro-nutrient consumption (column 1); how much of this effect comes from animals and how much from crops (column 2); and whether own production is crowding out market intake (by comparing results from columns 3 and 4). This is now achieved and various impact of FPD or its components is now well documented and the crowding effects also clearly elaborated.

Reviewer #2: This is a relevant manuscript that highlights the importance of producing diverse food products. My comments for consideration are below:

• More information is needed in the introduction section describing the common food production and consumption patterns in Uganda. This is now sufficiently done. What various types of crops and livestock are usually produced? This is also done. This is important because HDD score depends on food groups. Also, what proportion of farmers eat their own farm produce and what proportion do not. This is done. In other words, what proportion of own production is consumed? What types of farm products are usually sold for cash and not consumed by farmers? This is also sufficiently done.

• A few specific examples of crops considered in the counts would be useful since nutrient content and expected nutritional contributions may differ among crops. This is also done. Such information will be helpful in the discussions on page 17.

• The authors found a strong positive association of the crop species count with energy and micronutrients sourced from own farm produce and attributed this to farmers mostly consuming what they grow. So, my question is: what is the problem in Uganda? Farm production is still dominantly subsistence and perhaps insufficient to cater for all households’ nutritional and dietary needs. If farmers are producing and eating what they grow, then what is the gap? Why are they hungry or suffering from malnutrition? Largely grow cereals and tubers that mostly supply energy with minimal micronutrient contributions, and often times are produced in insufficient quantities that can’t fully satisfy all food and dietary needs of households. 

• In the last sentence on Page 18, what do you mean by essential foods? You listed cereals or their products, oils and fats etc. Why do you refer to these foods as essential? The word “essential” was perhaps used in a wrong context, this is clarified with better wording. Also, you need to distinguish between raw produce and processed products. Aren't the farmers producing cereals and foods rich in fats and oils e.g. groundnuts? Clarify this essential food category. This is well clarified now.

• You concluded that diversification in crop species could be more important than animal species diversification towards availing more energy, and micronutrients per adult. Were you expecting significant differences in nutritional contrition of different livestock species? What different species were counted? Unfortunately, we didn’t have fish production – but indeed the expectation would be valid that different animal species could contribute differently, just as is with crop species, say cereals versus pulses. Unfortunately, we don’t extend our analysis to specific species assessment for impact on specific nutrition outcomes. Perhaps, livestock and fisheries could have some differences in nutritional value and contributions to HDDS. If there is data on fish production and consumption, then it will be interesting and relevant to consider it in the analysis. we didn’t have data on fish production, and perhaps it happens only with large scale urban farmers who are missed in the UNPS general survey.

• Other comments: the manuscript needs to be better structured with introduction, methods and data analysis sections clearly separated. Last paragraph on page 4 – 5 could be moved to methods section. Information under your current methods section could be put under data analysis. This suggested re-arrangement of sections for a more logical flow, has been incorporated. 

Reviewer #3: The manuscript addresses an area of great importance to the success of policies, especially in resource-scarce settings. The manuscript is well-organised and easy to follow. I have read the manuscript and have made the following observations which I think the authors need to address before publication can be considered:

1. The manuscript requires significant proofreading, especially checking of punctuation. This has been done effectively, with more short sentences and single punctuation per sentence.

2. Section I: Introduction

a) Please justify why the study only investigated the impact of FPD on daily energy, iron, zinc, and vitamin A, but not any other nutrients. Particularly, how do these relate to the nutritional issues in Uganda? Selected micronutrients are justified in the introduction.

3. Section C: Data Description

a) It would be helpful for reader to give an overview of farm characteristics of the sample, e.g., location on map, location map is difficult because it considers all Uganda, so we feared it could add no value. names and types of crops cultivated this is clearly added, cropping system etc.

b) Referring to Table 1, the amount of energy and most micronutrients consumed from both market and own farm sources are far higher in year 2015 than in 2018 and 2019. Is there any explanation to this anomaly? This is observed but clear explanation a bit difficult to zero down too. Perhaps, could it have been due to larger land size and increasing land fragmentation in later years? We Checked-out rains data for 2015 versus other years and other specialties that could have favored more production, but this is available on regional level unlike particular locations for smallholder farmers locations. Could this anomaly affect the statistical rigour?

4. Methods, Results and Discussions

a) It appears that the volume of crop yield and its potential confounding effect on HDDS and nutritional outcome have not been considered or discussed sufficiently in the manuscript. Volume of crop yield couldn’t be considered in the model for fear of potential endogeneity with other X variables including FPD itself. Moreover, our outcome indicator – HDDS - is more of a quality indicator than a quantity one. Nevertheless, we acknowledge sufficiently now in our discussions that a limited volume of crop yield over the years could explain the minimal variation in nutrition outcomes over the years. 

b) “Each crop species added to those farmed with in a household was associated with 5.9 and 18.8 kilocalories (0.6 and 0.9 percentage points) added to energy sourced from own farm produce consumption or markets respectively. With regards to iron, each additional crop species was associated with 0.4 and 0.1 milligrams (4.2 and 0.7 percentage points) added to daily iron intake sourced from own farm produce or markets consumption respectively. With regards to zinc, each additional crop species grown on the farm was associated with 0.3 and 0.1 milligrams (6.2 and 0.9 percentage points) added to daily zinc intake sourced from own farm produce, and markets consumption respectively. Lastly, each additional crop species grown on farm was associated with 1.1 and 0.6 rae_micrograms (0.3 and 0.2 percentage points) added to daily vitamin A intake sourced from own farm produce, and markets consumption respectively.”

What is the significance of the percentages of gains quoted above in the diets of the sample population? Are these considered meaningful impact on nutritional status? Also, will increment in crop diversity continue to produce nutritional gains at a linear rate or will the effect level off? At what point will the effect level off? We added squared FPD to detect linear contributions, which are of course not continuous indefinitely, we discuss this in the paper. Check for the meaningful impact of changes by % in nutrition status. Percentages are just used to show relative proportions – however, we quote the exact thresholds for each micronutrient as advised by WHO and FAO, the percentages discussed in the paper, just help us to show the proportion of the effect, with regards to average values in the sample; which was earlier compared with thresholds to define energy and micronutrient deficient households and those that were not deficient.

c) “There are however other factors that are consistently and significantly associated with energy, and micronutrients intake, for instance gender effects, household size, and year variables, which we don’t discuss here to prioritize our focus on FPD sub-components, which are our main covariates.”

Wouldn’t understanding these other factors that are also consistently and significantly associated with energy, and micronutrients intake and the relationships of these factors with FPD help in devising more effective policy? Because we increased the coverage of the micronutrients and energy, accommodating each to own table, thus generating 4 tables from 1 original table – still we found ourselves limited by space. Therefore, we completely eliminated all other covariates from the main manuscript and thus their discussions but presented full tables in supporting information in Tables 3-6. 

5. Section VI: Policy Implications

a) This section is not sufficiently discussed. It is suggested that the government should devote more attention to crop species, but it does not recommend any specific measures. The scope of our analysis didn’t allow us to make specific crop recommendations as our analysis stopped on crops in general without specifications. The authors should focus on how to translate their research into practice at varying levels of scale, considering all other confounders and trade-offs. We clarify that finer guidance may need further research on particular crops or animal species that we didn’t consider in this study.

b) For various apparent reasons, the policymakers might be more inclined to invest in promotion of cash crops as a means to achieve rural development and poverty alleviation. Cash crops also encourages mono-cropping that reduces FPD. How can this dilemma be addressed? How can policy makers be convinced? Again, our focus didn’t disintegrate particular crops or animals – since there were already two levels of disintegration (at FPD level (animals or crops), and sources of considered nutrition outcomes (own farm or markets), thus further disintegration could be so much in one paper. But we refer to this as an interesting area of research that can be exploited in the future.

---

## [Decision Letter · Decision Letter 1]

5 Jul 2022

PONE-D-22-06649R1Farm Production Diversity, Household Dietary Diversity and Nutrition: Evidence from Uganda’s National Panel SurveyPLOS ONE

Dear Dr. Sekabira,

Thank you for submitting your manuscript to PLOS ONE. After careful consideration, we feel that it has merit but does not fully meet PLOS ONE’s publication criteria as it currently stands. Therefore, we invite you to submit a revised version of the manuscript that addresses the points raised during the review process.

 We have received comments from all 3 previous reviewers. Please note that Reviewer 3 has sent their comments in the document attached to this email. Please make sure you address all the concerns raised. 

We look forward to receiving your revised manuscript.

Kind regards,

Carla Pegoraro

Division Editor

PLOS ONE

Reviewers' comments:

Reviewer's Responses to Questions

**Comments to the Author**

1. If the authors have adequately addressed your comments raised in a previous round of review and you feel that this manuscript is now acceptable for publication, you may indicate that here to bypass the “Comments to the Author” section, enter your conflict of interest statement in the “Confidential to Editor” section, and submit your "Accept" recommendation.

Reviewer #1: (No Response)

Reviewer #2: All comments have been addressed

Reviewer #3: (No Response)

2. Is the manuscript technically sound, and do the data support the conclusions?

Reviewer #1: Partly

Reviewer #2: Yes

Reviewer #3: (No Response)

3. Has the statistical analysis been performed appropriately and rigorously? 

Reviewer #1: No

Reviewer #2: I Don't Know

Reviewer #3: (No Response)

4. Have the authors made all data underlying the findings in their manuscript fully available?

Reviewer #1: Yes

Reviewer #2: Yes

Reviewer #3: (No Response)

5. Is the manuscript presented in an intelligible fashion and written in standard English?

Reviewer #1: No

Reviewer #2: Yes

Reviewer #3: (No Response)

6. Review Comments to the Author

Reviewer #1: Review of: Farm Production Diversity, Household Dietary Diversity and Nutrition: Evidence from Uganda’s National Panel Survey (Manuscript Number: PONE-D-22-06649R1)

This revised paper examines associations between the diversity of crop and livestock products produced by households living in Uganda and measures of the diversity of their food consumption. Although there are already many papers published on this topic, this is a welcome addition to the literature as it includes new evidence from a country that previously had received relatively little attention. The authors are to be commended for taking seriously the concerns raised in my initial report and for extensively revising the paper.

That said, this revision has generated new concerns that were not apparent in the initial submission.

(1) The specification used in Table 2 includes linear and quadratic terms for measures of production diversity. Tables 3-6 use linear terms only. This does not make sense; you need to use the same specification for all tables.

(2) The quadratic formulation in Table 2 produces some very odd results. I calculated the marginal effects of increasing diversity by one unit, using the estimates reported in column (1). Going from 0 to 1 and from 1-2 units increases consumption diversity, but going from 2 to 3, 3 to 4 and 4 to 5 leads to reductions in consumption diversity. This makes no sense. The authors have a good idea, they want to allow for diminishing increases in consumption diversity as production diversity rises but a quadratic formulation is the wrong way of doing so. A much better approach would be to use an inverse hyperbolic sine (see Bellemare, Oxford Bulletin of Economics and Statistics); this allows for increases at a diminishing rate while also allowing for zero values in the independent variable (note that you will need to convert the parameter estimates to marginal effects; Bellemare provides the formulae for doing so).

(3) The introductory materials are far too long; these could be reduced in length by 50%.

(4) The paper needs to be copy-edited by a native English speaker; there are still many odd use of words and phrases (for example, the abstract states that nutrition outcomes are disintegrated by source when I think the authors mean to say “disaggregated by source.”

(5) As this is an observational study, any policy implications should be made very cautiously.

Reviewer #2: The authors have done well by addressing all my concerns. I find the manuscript suitable for publication.

Reviewer #3: (No Response)

7. PLOS authors have the option to publish the peer review history of their article (what does this mean?). If published, this will include your full peer review and any attached files.

Reviewer #1: No

Reviewer #2: No

Reviewer #3: **Yes: **Ee Von Goh

---

## [Author Response · Author response to Decision Letter 1]

2 Sep 2022

PONE-D-22-06649R1

Farm Production Diversity, Household Dietary Diversity and Nutrition: Evidence from Uganda’s National Panel Survey

PLOS ONE

CURRENT RESPONSES TO REVIEWERS’ COMMENTS ARE HIGHLIGHTED IN GREEN INK

Reviewer #1: 

That said, this revision has generated new concerns that were not apparent in the initial submission.

(1) The specification used in Table 2 includes linear and quadratic terms for measures of production diversity. Tables 3-6 use linear terms only. This does not make sense; you need to use the same specification for all tables.

We now drop the quadratic terms all through the specifications as advised by the reviewer, but consistently use the inverse hyperbolic sine of the respective FPD indices across all specifications, thus all tables.

(2) The quadratic formulation in Table 2 produces some very odd results. I calculated the marginal effects of increasing diversity by one unit, using the estimates reported in column (1). Going from 0 to 1 and from 1-2 units increases consumption diversity, but going from 2 to 3, 3 to 4 and 4 to 5 leads to reductions in consumption diversity. This makes no sense. The authors have a good idea, they want to allow for diminishing increases in consumption diversity as production diversity rises but a quadratic formulation is the wrong way of doing so. A much better approach would be to use an inverse hyperbolic sine (see Bellemare, Oxford Bulletin of Economics and Statistics); this allows for increases at a diminishing rate while also allowing for zero values in the independent variable (note that you will need to convert the parameter estimates to marginal effects; Bellemare provides the formulae for doing so).

We consider the reviewer’s advice and use the inverse hyperbolic sine of the respective FPD index all through the specifications – which now actually gives us a general view of significant and positive associations along the own-farm production consumption pathway, which is logical for totally rural smallholder households.

(3) The introductory materials are far too long; these could be reduced in length by 50%.

The introductory materials are re-written and reduced in volume with more concise and precise wording – maintaining the value of the content but eliminating the bulk.

(4) The paper needs to be copy-edited by a native English speaker; there are still many odd use of words and phrases (for example, the abstract states that nutrition outcomes are disintegrated by source when I think the authors mean to say “disaggregated by source.”

Indeed, this was a big oversight which we correct now – and rewrite the paper carefully with concise and straightforward words. We have also sought the assistance of a native English speaker to edit the paper.

(5) As this is an observational study, any policy implications should be made very cautiously.

Policy implications have been re-written with cautiously made recommendations that are only restricted to what the study covered.

Reviewer #2: 

The authors have done well by addressing all my concerns. I find the manuscript suitable for publication.

This is gracefully noted – and thankful for the reviewer’s confirmation.

Reviewer #3: 

I find the revision did not fully address my questions or answer. We now address more clearly the queries of the reviewer with more information on the same, only that this increases the bulk of the paper a bit

2. Section I: Introduction

a) Please justify why the study only investigated the impact of FPD on daily energy, iron,

zinc, and vitamin A, but not any other nutrients. Particularly, how do these relate to the

nutritional issues in Uganda? Selected micronutrients are justified in the introduction. We highlight the serious importance of the studied micronutrients to the growth and development of especially in children globally and in Uganda as well, showing how their deficiencies can lead to poor growth, death, and loss of economic productivity. We elaborate the justification of these micronutrients between lines 117 – 160 

The revised manuscript failed to answer why only energy, iron, zinc and vitamin A are selected. For instance:

- Energy intake does not appear to be a problem in the population, so why are you investigating the impact of FPD on daily energy? Even your results show that “Average consumption of energy (2,430 kilocalories) was slightly above FAO recommended thresholds per adult”. Because energy is the first general component needed for any organism to live, we saw it necessary to start our analyses with it before looking at micronutrients. Nevertheless, we justify why energy is important for life, and state that even though the average is slightly above the threshold, Uganda has been recently 2018 ranked 168th of 171 countries according to daily energy intake per capita. Therefore, consideration of energy in this regard was justifiable, at least to confirm that despite the low rank, the average is still above the threshold.

- As a side note, it is often more important to balance the macronutrients, i.e., the makeup of the calories one consumes is more important. In that light, what is the breakdown of calorie intake contributed by each class of macronutrients (carbs, proteins, fats)? Why is protein not investigated? Referring to “the regular Ugandan diet is majorly made up of starchy roots and tubers (cassava, sweet potatoes), plantains (mostly cooking bananas), and cereals (maize, millet, sorghum)”, these staples are low in protein, and since they form the bulk of the population’s daily diet, could the population be at risk of protein deficiency? Is stunted growth a problem in Uganda? The reviewer makes a very valid observation that indeed the population consumes fewer protein foods as would be expected of rural populations, and we agree that this is noble to study. Unfortunately, we are building on our 2021 work where in the original data, we did not compute for proteins. However, we make a stark recommendation to expand consideration of macro and micronutrient studies to other nutrients, even though such wider coverage may be difficult in one paper, but done individually for each macro or micronutrient. The other constraint is that we may not cover every macro and micronutrient study in one paper, but we try our best to give a fair picture that can be compared across micronutrients, without actually getting lost into so much detail.

- Iron – is iron-deficiency a public health concern in the population? What is the prevalence of iron-deficiency anaemia? 

- Same for zinc and vitamin A. Please justify why just these four nutrients are selected but not the rest. (Just because we can do something doesn't mean we should). Justification for all these micronutrients and their central importance in human growth is now considered in the paper, including local and country statistics that portray this importance. Because of such importance, we prioritized these micronutrients and the fact that our original data already was programmed to generate these micronutrients and not others. However, we acknowledge that it is also important to study other micronutrients if especially the data set-up and technical capabilities allow it.

3. Section C: Data Description

b) Referring to Table 1, the amount of energy and most micronutrients consumed from both

market and own farm sources are far higher in year 2015 than in 2018 and 2019. 

Could this anomaly affect the statistical rigour and interpretation? We re-look at the data (official results, our computations, and FOASTAT data to actually confirm that indeed production in 2015, was higher than any of the later years. However, we stress in our explanations on the same, that this time variations in production that could have stemmed from specific factors in specific years do not affect the rigor and interpretation of our results, since we use panel data models where we control for time fixed effects – to clean out any effects related to a particular year that weren’t normal in other years. We explain this in the data description section.

4. Methods, Results and Discussions

b) “Each crop species added to those farmed with in a household was associated with 5.9 and

18.8 kilocalories (0.6 and 0.9 percentage points) added to energy sourced from own farm produce consumption or markets respectively. With regards to iron, each additional crop

species was associated with 0.4 and 0.1 milligrams (4.2 and 0.7 percentage points) added to

daily iron intake sourced from own farm produce or markets consumption respectively. With

regards to zinc, each additional crop species grown on the farm was associated with 0.3 and

0.1 milligrams (6.2 and 0.9 percentage points) added to daily zinc intake sourced from own

farm produce, and markets consumption respectively. Lastly, each additional crop species

grown on farm was associated with 1.1 and 0.6 rae_micrograms (0.3 and 0.2 percentage

points) added to daily vitamin A intake sourced from own farm produce, and markets

consumption respectively.”

What is the significance of the percentages of gains quoted above in the diets of the sample

population? Are these considered meaningful impact on nutritional status? Also, will

increment in crop diversity continue to produce nutritional gains at a linear rate or will the

effect level off? At what point will the effect level off? We added squared FPD to detect

linear contributions, which are of course not continuous indefinitely, we discuss this in

the paper. Check for the meaningful impact of changes by % in nutrition status. Percentages

are just used to show relative proportions – however, we quote the exact thresholds for each

micronutrient as advised by WHO and FAO, the percentages discussed in the paper, just help

us to show the proportion of the effect, with regards to average values in the sample; which

was earlier compared with thresholds to define energy and micronutrient deficient households

and those that were not deficient. We now modify our specifications and use the inverse hyperbolic sine of FPD to show decreasing returns on nutrition indicators with an overly increased FPD.

Referring to my previous question in 2(a). If the “Average consumption of energy (2,430 kilocalories) was slightly above FAO recommended thresholds per adult”, why does “56

calories (2.3 percentage points) added for each species added to FPD” matter? This also relates to Policy Implications. We use these percentage points metrics to show a comparable size of the increment or the decrease. It helps policy makers to see where larger returns can be attained per unit of investment (crop or livestock species) farmed. Therefore, with minimal resources these comparative results can guide policy on which areas to prioritize among all analyzed aspects, for a choice to attain maximum possible impact per unit of investment. We make some justification for studying energy even when it was generally above recommended thresholds. Recently, in 2018, Uganda has been ranked 168th out of 171 countries, indicating that generally Uganda wasn’t fairing well visa viz other countries. Also, energy is the first basic requirement to ensure life, so this also made it important that we looked at it, especially when our data programming could allow it.

---

## [Decision Letter · Decision Letter 2]

26 Oct 2022

PONE-D-22-06649R2Farm Production Diversity, Household Dietary Diversity and Nutrition: Evidence from Uganda’s National Panel SurveyPLOS ONE

Dear Dr. Sekabira,

Thank you for submitting your manuscript to PLOS ONE. After careful consideration, we feel that it has merit but does not fully meet PLOS ONE’s publication criteria as it currently stands. Therefore, we invite you to submit a revised version of the manuscript that addresses the points raised during the review process. There are still some minor issues to solve: the main regards the functional specification used in the estimates. As one of the reviewer indicated, you should include only the IHS transforms of crop and animal diversity without the linear term. Please, calculate also the marginal effects from parameter estimates following the formulas provided by Bellemare  (Oxford Bulletin of Economics and Statistics).

We look forward to receiving your revised manuscript.

Kind regards,

Francesco Caracciolo

Academic Editor

PLOS ONE

Journal Requirements:

Reviewers' comments:

Reviewer's Responses to Questions

**Comments to the Author**

1. If the authors have adequately addressed your comments raised in a previous round of review and you feel that this manuscript is now acceptable for publication, you may indicate that here to bypass the “Comments to the Author” section, enter your conflict of interest statement in the “Confidential to Editor” section, and submit your "Accept" recommendation.

Reviewer #1: (No Response)

Reviewer #2: All comments have been addressed

Reviewer #3: (No Response)

2. Is the manuscript technically sound, and do the data support the conclusions?

Reviewer #1: No

Reviewer #2: Yes

Reviewer #3: Yes

3. Has the statistical analysis been performed appropriately and rigorously? 

Reviewer #1: No

Reviewer #2: Yes

Reviewer #3: Yes

4. Have the authors made all data underlying the findings in their manuscript fully available?

Reviewer #1: No

Reviewer #2: Yes

Reviewer #3: (No Response)

5. Is the manuscript presented in an intelligible fashion and written in standard English?

Reviewer #1: Yes

Reviewer #2: Yes

Reviewer #3: Yes

6. Review Comments to the Author

Reviewer #1: Review of: Farm Production Diversity, Household Dietary Diversity and Nutrition: Evidence from Uganda’s National Panel Survey (Manuscript Number: PONE-D-22-06649R2)

As noted in an earlier report, this paper examines associations between the diversity of crop and livestock products produced by households living in Uganda and measures of the diversity of their food consumption. This is a welcome addition to the literature as it includes new evidence from a country that previously had received relatively little attention. And again, the authors are to be commended for taking seriously the concerns raised in my second report and for extensively revising the paper.

Unfortunately, however, they have misunderstood a crucial comment that I had made. The issue is this. They seek to look at the associations between crop and animal diversity in production and measures of food consumption and food consumption diversity. They want to allow this relationship to be non-linear. However, they cannot use a quadratic specification because the second order term may have a negative coefficient, implying that at some point increased production diversity lowers consumption diversity. They cannot use a logarithmic specification because they may have zero values for their measures of production diversity.

The solution to this problem is to use an inverse hyperbolic sine (see Bellemare, Oxford Bulletin of Economics and Statistics); this allows for increases at a diminishing rate while also allowing for zero values in the independent variable (note that you will need to convert the parameter estimates to marginal effects; Bellemare provides the formulae for doing so). The authors include this in their model but also include a linear term. It does not make sense to include both. The models reported in Tables 2-6 should only include the IHS transforms of crop and animal diversity as well as other control variables.

Reviewer #2: (No Response)

Reviewer #3: 1. Justifications for indicators selection (lines 95 – 160) are a little too long-winded. Although some redundancy can be useful, this section would be greatly improved by making it a lot more concise.

2. Re: "The reviewer makes a very valid observation that indeed the population consumes fewer protein foods as would be expected of rural populations, and we agree that this is noble to study. Unfortunately, we are building on our 2021 work where in the original data, we did not compute for proteins. However, we make a stark recommendation to expand consideration of macro and micronutrient studies to other nutrients, even though such wider coverage may be difficult in one paper, but done individually for each macro or micronutrient. The other constraint is that we may not cover every macro and micronutrient study in one paper, but we try our best to give a fair picture that can be compared across micronutrients, without actually getting lost into so much detail."

Yes, it would be useful to acknowledge the shortcoming and why you thought studying protein (and other micronutrients) in future research is important. But I don't seem to be able to locate these in the manuscript.

7. PLOS authors have the option to publish the peer review history of their article (what does this mean?). If published, this will include your full peer review and any attached files.

Reviewer #1: No

Reviewer #2: No

Reviewer #3: No

---

## [Author Response · Author response to Decision Letter 2]

9 Nov 2022

PONE-D-22-06649R2

Farm Production Diversity, Household Dietary Diversity and Nutrition: Evidence from Uganda’s National Panel Survey

PLOS ONE

Our responses to the editor and reviewers’ comments/suggestions are marked in red

Editors’ comments

There are still some minor issues to solve: the main one regards the functional specification used in the estimates. As one of the reviewers indicated, you should include only the IHS transforms of crop and animal diversity without the linear term. Please, calculate also the marginal effects from parameter estimates following the formulas provided by Bellemare (Oxford Bulletin of Economics and Statistics). 

This is noted and has been extensively worked upon, as exactly the reviewer guided.

•A rebuttal letter that responds to each point raised by the academic editor and reviewer(s). You should upload this letter as a separate file labeled 'Response to Reviewers'. 

This file has been constructed and uploaded in the revisions.

•A marked-up copy of your manuscript that highlights changes made to the original version. You should upload this as a separate file labeled 'Revised Manuscript with Track Changes'. This file has been constructed and uploaded in the revisions.

•An unmarked version of your revised paper without tracked changes. You should upload this as a separate file labeled 'Manuscript'. 

This file has been constructed and uploaded in the revisions.

No changes made.

Please review your reference list to ensure that it is complete and correct. If you have cited papers that have been retracted, please include the rationale for doing so in the manuscript text or remove these references and replace them with relevant current references. Any changes to the reference list should be mentioned in the rebuttal letter that accompanies your revised manuscript. If you need to cite a retracted article, indicate the article’s retracted status in the References list and also include a citation and full reference for the retraction notice. 

Two references of Bellemare and Wichman (2020) and Bellemare et al. (2013) – with empirical guidance on IHS implementation and usage have been added.

Reviewers' comments:

Reviewer #1: 

As noted in an earlier report, this paper examines associations between the diversity of crop and livestock products produced by households living in Uganda and measures of the diversity of their food consumption. This is a welcome addition to the literature as it includes new evidence from a country that previously had received relatively little attention. And again, the authors are to be commended for taking seriously the concerns raised in my second report and for extensively revising the paper. 

We are delighted with the observation and comment from the reviewer.

Unfortunately, however, they have misunderstood a crucial comment that I had made. The issue is this. They seek to look at the associations between crop and animal diversity in production and measures of food consumption and food consumption diversity. They want to allow this relationship to be non-linear. However, they cannot use a quadratic specification because the second order term may have a negative coefficient, implying that at some point increased production diversity lowers consumption diversity. They cannot use a logarithmic specification because they may have zero values for their measures of production diversity. 

This guidance is appreciated and plausible and has been incorporated as earlier advised by the reviewer. 

The solution to this problem is to use an inverse hyperbolic sine (see Bellemare, Oxford Bulletin of Economics and Statistics); this allows for increases at a diminishing rate while also allowing for zero values in the independent variable (note that you will need to convert the parameter estimates to marginal effects; Bellemare provides the formulae for doing so). The authors include this in their model but also include a linear term. It does not make sense to include both. The models reported in Tables 2-6 should only include the IHS transforms of crop and animal diversity as well as other control variables. 

This guiding solution has now been extensively implemented as guided by the reviewer, following Bellemare and Wichman 2020, hopefully to the satisfaction of the reviewer.

Reviewer #2: 

(No Response) 

This is noted, and we are delighted to have satisfied the reviewer.

Reviewer #3: 

1. Justifications for indicators selection (lines 95 – 160) are a little too long-winded. Although some redundancy can be useful, this section would be greatly improved by making it a lot more concise. 

Guidance is noted, and the section is shortened more concisely.

2. Re: "The reviewer makes a very valid observation that indeed the population consumes fewer protein foods as would be expected of rural populations, and we agree that this is noble to study. Unfortunately, we are building on our 2021 work where in the original data, we did not compute for proteins. However, we make a stark recommendation to expand consideration of macro and micronutrient studies to other nutrients, even though such wider coverage may be difficult in one paper, but done individually for each macro or micronutrient. The other constraint is that we may not cover every macro and micronutrient study in one paper, but we try our best to give a fair picture that can be compared across micronutrients, without actually getting lost into so much detail."

Yes, it would be useful to acknowledge the shortcoming and why you thought studying protein (and other micronutrients) in future research is important. But I don't seem to be able to locate these in the manuscript. 

Indeed, as observed by the reviewer, this was not included in the manuscript, but we now concisely make mention of it. We are grateful for this advice.

---

## [Decision Letter · Decision Letter 3]

6 Dec 2022

Farm Production Diversity, Household Dietary Diversity and Nutrition: Evidence from Uganda’s National Panel Survey

PONE-D-22-06649R3

Dear Dr. Sekabira,

We’re pleased to inform you that your manuscript has been judged scientifically suitable for publication and will be formally accepted for publication once it meets all outstanding technical requirements.

Kind regards,

Francesco Caracciolo

Academic Editor

PLOS ONE

Additional Editor Comments (optional):

Reviewers' comments:

Reviewer's Responses to Questions

**Comments to the Author**

1. If the authors have adequately addressed your comments raised in a previous round of review and you feel that this manuscript is now acceptable for publication, you may indicate that here to bypass the “Comments to the Author” section, enter your conflict of interest statement in the “Confidential to Editor” section, and submit your "Accept" recommendation.

Reviewer #1: All comments have been addressed

2. Is the manuscript technically sound, and do the data support the conclusions?

Reviewer #1: Yes

3. Has the statistical analysis been performed appropriately and rigorously? 

Reviewer #1: Yes

4. Have the authors made all data underlying the findings in their manuscript fully available?

Reviewer #1: Yes

5. Is the manuscript presented in an intelligible fashion and written in standard English?

Reviewer #1: Yes

6. Review Comments to the Author

Reviewer #1: (No Response)

7. PLOS authors have the option to publish the peer review history of their article (what does this mean?). If published, this will include your full peer review and any attached files.

Reviewer #1: No

---

## [Editor Report · Acceptance letter]

8 Dec 2022

PONE-D-22-06649R3 

Farm production diversity, household dietary diversity, and nutrition: Evidence from Uganda’s national panel survey 

Dear Dr. Sekabira:

I'm pleased to inform you that your manuscript has been deemed suitable for publication in PLOS ONE. Congratulations! Your manuscript is now with our production department. 

Kind regards, 

on behalf of

Dr. Francesco Caracciolo 

Academic Editor

PLOS ONE